# Regulation of chromatin modifications through coordination of nucleus size and epithelial cell morphology heterogeneity
Alexandra Bermudez[1], Zoe D. Latham[1], Alex J. Ma[1], Dapeng Bi [2], Jimmy K. Hu [3,4,5] ✉ &
Neil Y. C. Lin [1,5,6,7,8] ✉

Cell morphology heterogeneity is pervasive in epithelial collectives, yet the underlying mechanisms driving such heterogeneity and its consequential biological ramifications remain elusive. Here, we observed a consistent correlation between the epithelial cell morphology and nucleus morphology during crowding, revealing a persistent log-normal probability distribution characterizing both cell and nucleus areas across diverse epithelial model systems. We showed that this morphological diversity arises from asymmetric partitioning during cell division. Next, we provide insights into the impact of nucleus morphology on chromatin modifications. We demonstrated that constraining nucleus leads to downregulation of the euchromatic mark H3K9ac and upregulation of the heterochromatic mark H3K27me3. Furthermore, we showed that nucleus size regulates H3K27me3 levels through histone demethylase UTX. These findings highlight the significance of cell morphology heterogeneity as a driver of chromatin state diversity, shaping functional variability within epithelial tissues.

Variability is inherent in biological systems and can influence the regulations and outputs of biological processes at all length scales[1,2]. At the molecular level, chemical reaction noise leads to uncertainties in transcription[3], translation[4], and post-translational modifications[5]. At the cellular level, stochastic gene expression imposes profound influences on cell functions, including cell cycle progression[6], metabolism[7], and stress response[8,9]. Over the past few decades, substantial efforts have been made to understand the role of noise in gene expression and genome organization, which are usually considered as "upstream noise"[10,11]. In contrast, cell-to-cell phenotypic heterogeneity, such as morphological variability, is viewed as a consequence of genetic and environmental variations[12,13]. As a result, the emergence of cell morphological heterogeneity within a population and its role in regulating gene expression has been relatively underexplored.

Morphological heterogeneity is one of the most noticeable features of phenotypic variability in epithelial cells. Even in a clonal population where all cells are genetically identical, each cell can exhibit a unique size and aspect ratio (AR)[14]. Specifically, during epithelial cell crowding, where cells continue to proliferate until a tightly packed cell layer is formed, each cell's shape can be controlled by its cytoskeletal properties and interactions with the neighboring cells and substrate[15,16]. In addition to these biological

regulations, recent studies have also found that cell shape AR in a population follows a nearly universal distribution across different organisms and systems[17,18]. This universal distribution can be explained by physical principles, such as the packing constraint[17], fundamental polygon geometric properties[18], and topological optimal transport[19]. These findings thus suggest that the statistical properties of epithelial cell shapes can be governed by mechanisms that transcend molecular details. However, the process through which morphological heterogeneities initially arise during crowding is not clear, and how similar mechanisms may also regulate the morphological heterogeneity in other organelles, such as the nucleus, remains an important open question.

The functional significance of cell shapes are highlighted in several recent studies that demonstrated the impact of cell shapes on chromatin structures and gene expression[20,21]. Leveraging engineered single-cell systems, these studies have uncovered the link between cell shapes and chromatin states. For example, cell shapes altered by micropatterns can directly influence the levels of lysine trimethylation[22] and acetylation[20] of histones, as well as influence chromosome positioning[23] and chromatin dynamics[21]. In addition, deformation of nucleus and cell shapes using topological micropillars can induce chromatin structure and gene expression changes[24,25].

[1]Bioengineering Department, University of California Los Angeles, Los Angeles, CA, USA. [2]Department of Physics, Northeastern University, Boston, MA, USA. [3]School of Dentistry, University of California Los Angeles, Los Angeles, CA, USA. [4]Molecular Biology Institute, University of California Los Angeles, Los Angeles, CA, USA. [5]Broad Stem Cell Center, University of California Los Angeles, Los Angeles, CA, USA. [6]Mechanical and Aerospace Engineering Department, University of California Los Angeles, Los Angeles, CA, USA. [7]Jonsson Comprehensive Cancer Center, University of California Los Angeles, Los Angeles, CA, USA. [8]Institute for Quantitative and Computational Biosciences, University of California Los Angeles, Los Angeles, CA, USA. ✉e-mail: jkhu@g.ucla.edu; neillin@g.ucla.edu

However, how the epigenetic states of cells are regulated by cell morphological variations in a physiologically relevant setting is not well understood.

To address these questions, we base our analyses on collective cell studies during epithelial crowding, which emulates physiological processes of epithelial biology and has been widely used for studying the functional roles of physical cues[26,27]. In this study, we focus on the emergence of morphological variations and their roles in determining chromatin modifications and organizations. Specifically, we characterized both the cellular and nuclear sizes, as the nucleus contains key architectural structures that interact with the genome[28,29] and has been shown to be an important organelle in mechanotransduction[30,31]. By combining such quantitative measurements with pharmacological perturbations, we show that cell and nucleus sizes follow a common log-normal probability distribution, and the cell-nuclear size correlation is regulated by both actomyosin tension and intracellular osmotic pressure balance. Importantly, cell and nucleus size variations are established and maintained with each cell doubling event, and these size differences influence the expression of UTX, a histone demethylase, which in turn modulates chromatin methylation states. Nuclear constraint thus reduces UTX levels, leading to increased H3K27me3. Our results demonstrate that cell morphological heterogeneity is not merely a noise during epithelial crowding, but has functional implications in directing nucleus sizes and subsequently modulating chromatin changes.

## Results

### Correlated evolution of cell and nucleus morphology throughout cell crowding

To understand how cell morphology affects cell behavior, we first set out to examine the evolution of both cell and nucleus sizes, which have been shown to mediate mechanical and geometric cues[30]. To do so, we cultured Madin Darby Canine Kidney (MDCK) cells with a seeding density of 30k cells/cm$^2$ and conducted cell and nucleus segmentation at 24, 64, 72, and 104 h after seeding (Fig. 1A). These timepoints were chosen to capture critical transitions in cell confluency and behavior, encompassing subconfluence, confluence, crowding onset, and a crowded steady-state[32]. These distinct stages capture the evolution of cell-substrate and cell-cell interactions, and changes in adhesion strengths and motilities[33]. Following known hallmarks of crowding, cell area and nucleus area became progressively smaller, while the adherence junction protein E-cadherin (E-cad) became upregulated and more localized to the intercellular junction[34].

We next quantified the cell and the nucleus size changes across timepoints. While nucleus and cell sizes refer to their three-dimensional (3D) volumes, within a monolayer, their projected areas are well accepted to approximate the cell and nucleus sizes[35,36]. Measuring areas has the benefit of improved statistics, as it enables a higher throughput of data acquisition and analysis. Studying cell and nucleus area would also allow us to compare results with previous studies that mainly performed two-dimensional (2D) analyses[37]. To validate using area as an approximation for volume in our system, we analyzed the nucleus volume using 3D image stacks (Fig. S1A) and showed a strong correlation between area and volume (Pearson correlation coefficient ~0.81) in a confluent cell layer (Fig. S1B). We further validated our size approximation by demonstrating that the cell layer is relatively flat (Fig. 1A middle row and Fig. S1C) throughout crowding, consistent with previous findings[38]. Together, these results confirmed the reduction of cell and nuclear sizes during epithelial crowding (Fig. 1B, C). Importantly, we found that both cells and nuclei exhibited morphological heterogeneity throughout crowding. The magnitude of such morphological variability, denoted by the large whiskers shown in Fig. 1B, C, well surpassed the magnitude of overall decrease in cell and nucleus area. This finding suggests that the intrapopulation cell-cell variability should not be overlooked when assessing individual cell behavior, as interpreting cell properties solely based on the global average may gloss over important biological information[39].

While the variabilities in both cell and nucleus areas may be initially regarded as biological noise, we identified a significant correlation between these characteristics (Fig. 1D). Specifically, we found that the nucleus and cell areas are positively correlated throughout crowding, indicating a constant nucleus area to cell area ratio (NC ratio) at each timepoint. Furthermore, this ratio becomes constant by the 64 hr timepoint when the cell layer reaches confluence. The lower NC ratio at the 24 h timepoint may be due to cell spreading associated with higher traction forces and actomyosin tension in the subconfluent state when compared to confluent cells[40]. To further test if NC ratio is also correlated in other epithelial model systems, we examined the human keratinocyte cell line (HaCaT) and the developing mouse epithelium at embryonic day (E) 12.5, in which the scatter plots are shown in Fig. S2. In both cases, we observed NC correlations. Here, while the mouse epithelium is a 3D tissue, we analyzed the outermost cells, which form a differentiated and flat layer that can be considered as a 2D model system[41]. The consistent observation of NC correlation in different models suggests that the nucleus-cell area co-regulation is conserved and likely regulated by a common mechanism across systems.

The observed ubiquitous NC correlation suggests that the morphological variability of cell and nucleus could be governed by a coordinated process throughout the highly dynamic cell crowding process. To understand the nature of this NC correlation, we next assessed the statistical commonality of such a morphological variability by normalizing the cell and nucleus areas to their respective means (Fig. 1E). Aside from the MDCK 24 hr cell area, this normalization revealed a universal collapse of all probability distribution function (PDF) curves, indicating that the area variability for both nucleus and cell share the same statistical properties. We further found that these distributions can be described by a log-normal fit $\text{PDF}(x) \sim \frac{0.25}{x} \exp(-\frac{(\ln x + 0.05)^2}{0.19})$, consistent with previous findings[34]. Also, the collapse of all curves demonstrates that the degree of area heterogeneity is independent of the post-confluent cell density and model systems, suggesting that the observed feature is conserved in human cells and developing mouse tissues.

Another morphological hallmark of crowding is the transition from an elongated to rounded cell shape, which we confirmed in our MDCK system by measuring the cell AR across different timepoints (Fig. 1F). Similar to the area analysis, we also measured nucleus AR throughout crowding (Fig. 1G) and found that as the cell AR decreased by 20%, the nucleus AR increased by 10% from the 24 h timepoint to the 104 hr timepoint. Moreover, both cell and nucleus ARs exhibited variabilities that are greater than the mean change over time (~1.5 × mean). Consistent with our area measurements, this observation suggests that heterogeneity must be considered to understand differences between individual cells.

Similar to the nucleus-cell area correlation, there is a mild, but statistically significant, correlation between cell and nucleus ARs in MDCK cells (Fig. 1H), HaCaT cells (Fig. S3A), and the mouse epithelium (Fig. S3B). The ratio between the nucleus and cell ARs increases during crowding, reflecting the simultaneous decrease in cell AR (Fig. 1F) and increase in nucleus AR (Fig. 1G). After scaling the AR distribution using a previously published[17] form $x = (AR - 1)/(\langle AR \rangle - 1)$ where $\langle \rangle$ denotes average, we found that all normalized AR PDFs collapsed to a common curve that can be described by a Gamma distribution $\text{PDF}(x; k) = k^k x^{k-1} e^{-kx}/\Gamma(k)$, where $\Gamma(k)$ is the Legendre gamma function with $k \sim 2.43$ for our best fit (Fig. 1I). This finding is consistent with a previous study suggesting that AR heterogeneity in crowded epithelia may be universally described by geometric constraints during jamming transition[17]. Notably, we found the correlation between area and AR negligible, in which the mean Pearson correlation coefficient is ~0.05 (Fig. S4). This finding suggests that these two variables are independent morphological features and governed by distinct mechanisms. Overall, our results illustrate that both cells and nuclei share common statistical properties of morphological variability across three distinct epithelial cell models.

### Size heterogeneity emerges following cell division

The overall reduction in cell area during crowding must arise from the production of new cells through cell proliferation within a limited space. It is also well known that cell division plays a key role in inducing morphological changes from confluence to a crowded state[17]. In our experiment, we

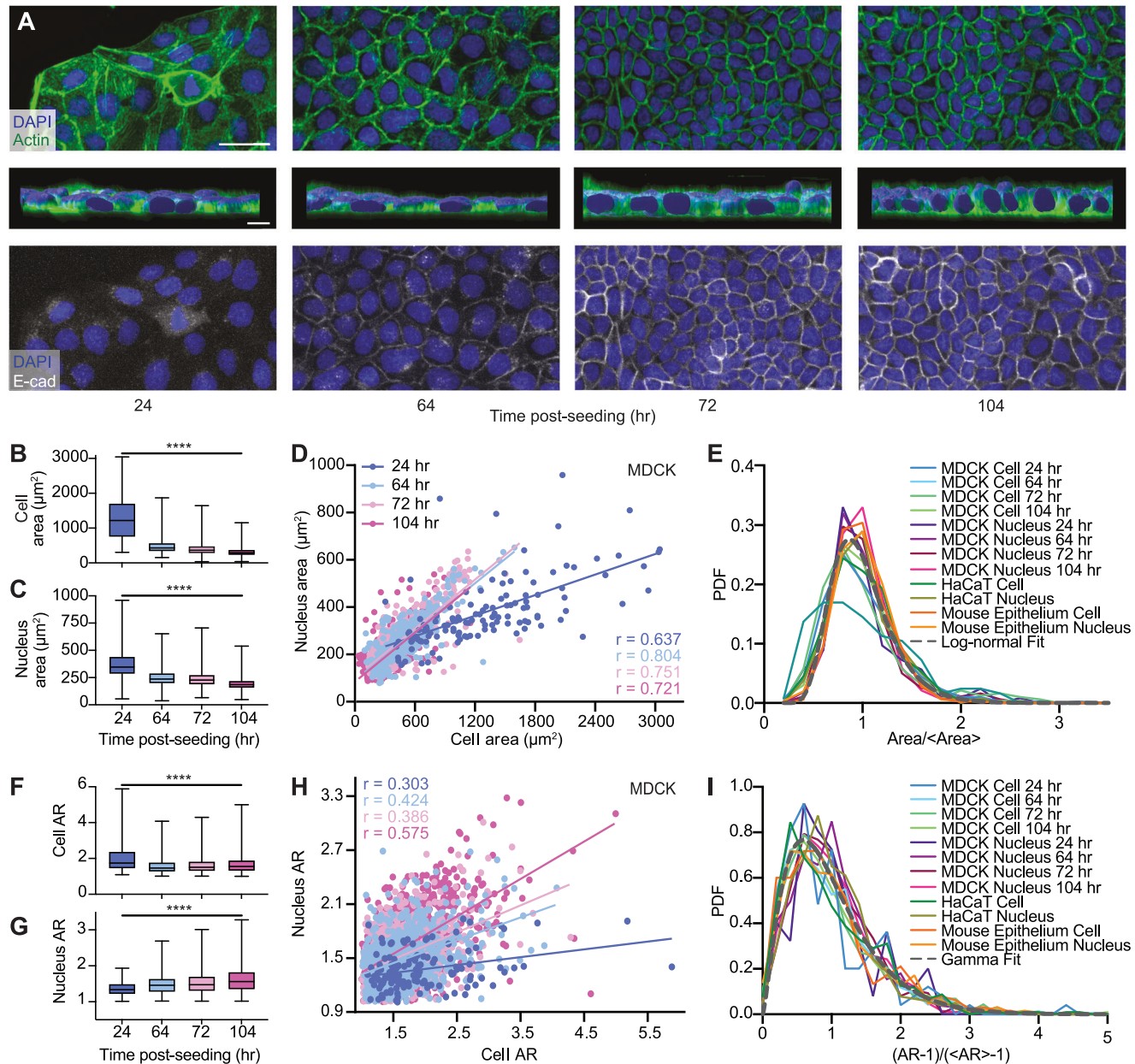

**Fig. 1 | Coordinated morphological changes in cells and nuclei during crowding.**
**A** Images of MDCK cells demonstrating evolution in cell and nucleus morphology
from being subconfluent to crowded (top). Cross-sectional 3D reconstruction of
MDCK cells demonstrating monolayer flatness throughout crowding (middle).
Scale bar = 10 μm. DAPI and actin staining shows that cells progressively acquire a
cobblestone-like morphology with decreasing sizes of both cell and nucleus during
crowding. E-cadherin (E-cad) staining illustrates the maturation of intercellular
junctions in late crowding. Scale bar = 50 μm. **B** Quantification of cell area
throughout crowding illustrates that the cell area variability surpasses the mean
change. $N$ = 124, 732, 803, and 1033 for 24, 64, 72, and 104 h analyses, respectively.
**C** Quantification of nucleus area throughout crowding illustrates that the nucleus
area variability also surpasses the mean change. $N$ = 147, 837, 840, and 1107 for 24,
64, 72, and 104 h analyses, respectively. **D** Persisting nucleus-cell area correlation
throughout crowding. 64 h, 72 h, 104 h datasets exhibit the same NC ratio, indicated
by the same slope of the best fits (solid lines). $N$ = 124, 808, 802, and 1033 for 24, 64,
72, and 104 h analyses, respectively. Solid lines represent best linear fits. $p < 0.0001$
for all timepoints. 95% confidence intervals corresponding to the 24, 64, 72, and
104 h data are [0.518, 0.731], [0.810, 0.853], [0.771, 0.822], and [0.668, 0.731],

respectively. Correlation coefficients corresponding to the 24, 64, 72, and 104 h data
are 0.637, 0.804, 0.751, and 0.721, respectively. **E** Normalized probability density
functions (PDF) for MDCK, HaCaT, and developing E12.5 mouse epithelium cell
and nucleus area. All PDFs, except the MDCK cell 24 h PDF, collapse on a master
curve and can be described by a log-normal fit. **F** Quantification of cell aspect ratio
(AR) throughout crowding. $N$ = 124, 732, 803, and 1033 for 24, 64, 72, and 104 h
analyses, respectively. **G** Quantification of nucleus AR throughout crowding.
$N$ = 124, 808, 803, and 1033 for 24, 64, 72, and 104 h analyses, respectively.
**H** Nucleus-cell AR correlation during crowding showing progressively increased
slopes over time. Solid lines represent best linear fits (solid lines). $N$ = 124, 808, 803,
and 1033 for 24, 64, 72, and 104 h analyses, respectively. $p < 0.0001$ for all time-
points. 95% confidence intervals corresponding to the 24, 64, 72, and 104 h data are
[0.134, 0.455], [0.366, 0.479], [0.388, 0.499], and [0.533, 0.614], respectively. Cor-
relation coefficients corresponding to the 24, 64, 72, and 104 h data are 0.303, 0.424,
0.386, and 0.575, respectively. **I** Normalized PDFs for MDCK, HaCaT, and mouse
epithelium cell and nucleus AR collapse on a master curve, which can be described by
a gamma distribution. **** refers to $p < 0.0001$. 3 biological replicates were used for
all analyses.

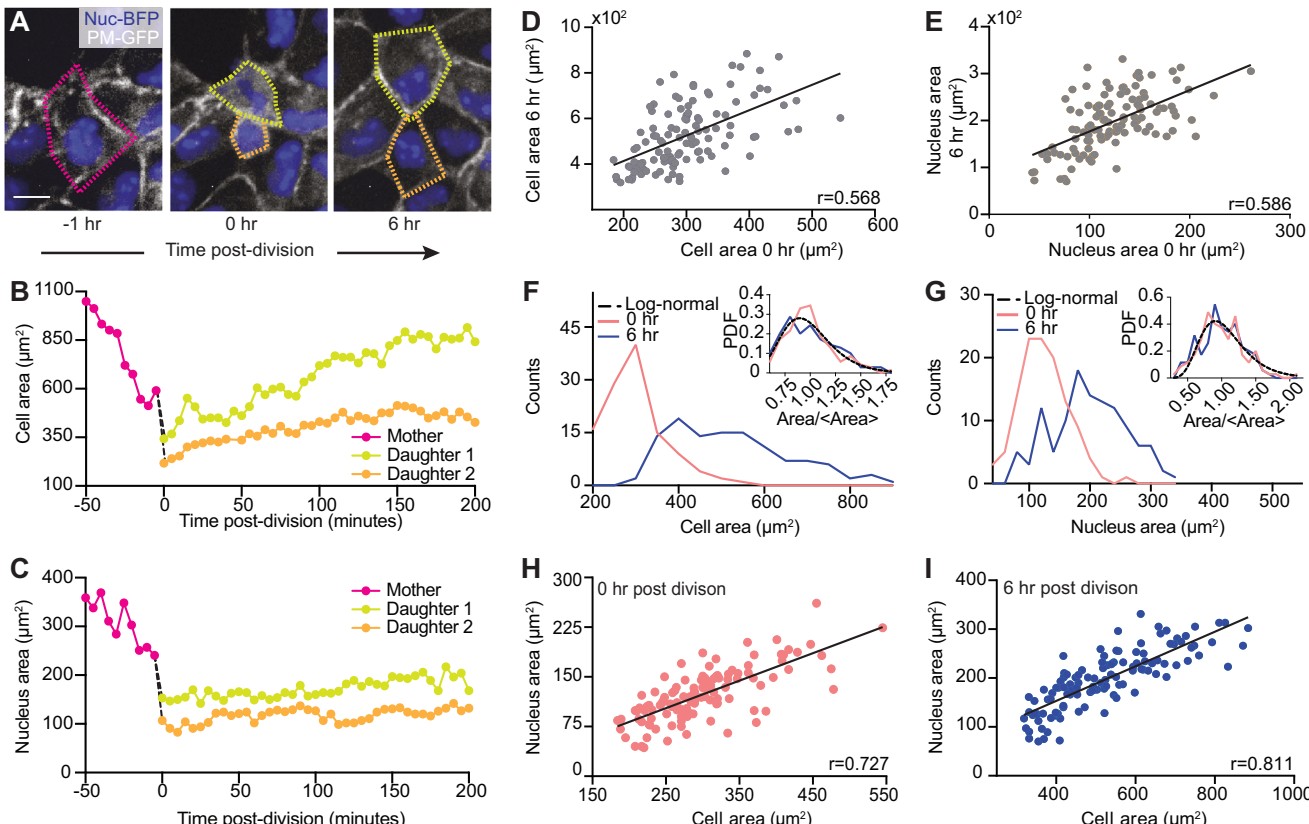

**Fig. 2 | Variations in cell and nucleus morphologies arise from uneven cell division. A** Images of live cells illustrating cell division and growth. A mother cell 1 h before division (left). Fuchsia outline denotes mother cell contour. Daughter cells possess different sizes immediately upon division (middle). Yellow outline denotes the larger daughter cell while orange outline denotes the smaller daughter cell. Daughter cells 6 hr after division (right). Scale bar = 15 μm. **B** Cell area evolution of mother and daughter cells shown in (**A**) illustrates that the area difference increased during cell growth. **C** Corresponding nucleus size evolution shows that the nucleus area differences remained during cell growth. **D** Cell areas at 0 and 6 h after division are correlated. $N = 116$. $p < 0.0001$. 95% confidence interval is [0.431, 0.679]. **E** Nucleus areas at 0 and 6 h after division are correlated. $N = 116$. $p < 0.0001$. 95% confidence interval is [0.452, 0.694]. **F** Histogram of cell areas 0 and 6 h after division. Inset shows that the normalized cell area PDFs of the 0 h and 6 h data can be described by the same log-normal distribution (black dashed line). **G** Corresponding histogram of nucleus areas 0 and 6 h after division. Inset shows that the normalized nucleus area PDFs of the 0 h and 6 h data can be described by the same log-normal distribution (black dashed line). **H** Nucleus-cell area correlation immediately upon division. $N = 116$. $p < 0.0001$. 95% confidence interval is [0.628, 0.803]. **I** Nucleus-cell area correlation 6 h after division. Black lines in (**D**, **E**, **H**, **I**) represent best linear fits. $N = 116$. $p < 0.0001$. 95% confidence interval is [0.738, 0.865]. 3 biological replicates were used for all analyses.

observed that the cell number tripled from the 64 h to 104 h timepoints (Fig. S5). This cell number increase prompted us to hypothesize that cell and nuclear morphological heterogeneity may arise following cell divisions, at which point the regulation of NC ratio may have also begun. We therefore set out to live-track the cellular and nuclear area of dividing cells and their daughter cells, using cells with the plasma membrane labeled with green fluorescent protein (GFP) and the nucleus labeled with blue fluorescent protein (BFP) (Fig. 2A). We focused our analysis using ~95%-confluent monolayers, where cells were crowded but remained proliferative (Fig. S5). We then measured the cellular and nuclear area of dividing cells and their daughter cells for 6 h to observe their morphological evolutions during cell growth, at which point a stable size is reached (Fig. S6).

As shown by the evolution of cell areas, we found that daughter cells from the same mother do not possess equal areas (0 h in Fig. 2B), rather one is larger than the other. Despite having identical genomes, there is a corresponding difference in nucleus area, with the larger daughter cell owning a larger nucleus (0 h in Fig. 2C). While the size differences between daughter cell size increased, the differences between their nuclei remained relatively constant after ~3 h of growth. By analyzing 58 pairs of daughters, we confirmed that the area disparity between the two daughters persisted as the cells grew, as demonstrated by the positive correlation between the 0 h and 6 h areas for both the cell (Fig. 2D) and the nucleus (Fig. 2E). The scatter plots also illustrate that the cell area increased more, with a slope of ~1.12,

than the nucleus area, with a slope of ~0.88, revealing that although co-regulated, most growth occurs within the cells and less so in their nuclei.

The persisting area difference leads to the question of whether the area variability observed in the steady-state system could simply originate from uneven cell divisions. To assess this possibility, we analyzed the statistical properties of cell area and nucleus area at 0 h and 6 h after division by plotting their PDFs in Fig. 2F, G, respectively. Consistent with Fig. 2D, E, we found that the distributions of both cell and nucleus area simply broadened as the cells grew. Notably, all PDFs were unimodal and skewed. We found that after normalization, both cell (Fig. 2F, inset) and nucleus (Fig. 2G, inset) area PDFs can be described by a universal log-normal distribution. This finding suggests that the area variability emerged from uneven cell divisions and was approximately linearly amplified during cell growth. To understand whether this randomness dominates over any inherited lineage dependent trait, we calculated the cell area-area autocorrelation function, which decays by an order of magnitude at the nearest neighbor distance of ~10 μm (Fig. S7). Such a short correlation length suggests that the final daughter cell area is not an inherit property from mother cells.

The similar evolution between cell and nucleus areas during cell growth indicates that the NC ratio regulation begins as soon as the cell division is completed. To test this, we studied the NC ratio correlation at 0 h (Fig. 2H) and 6 h (Fig. 2I) after division. While we observed a strong correlation regardless of the time after division, the ratio was higher at the 0 h than at the

6 h timepoint, which can be attributed to the increase in cell size due to subsequent cell growth after division. This finding indicates that the correlation between cell and nucleus is established before cell growth reaches a steady state following division, and such a correlation is maintained despite their disproportionate size increase during cell growth.

Collectively, our cell division tracking experiments suggest that the cell and nucleus area variability is established in two sequential steps. The first step involves uneven cell division, similar to previously established bacteria models[42]. In such models, it has been shown that cell division involves a series of multiplicative random events, which lead to a log-normal distribution of cell size, consistent with our measurements. The second step in the establishment of area distribution is the variability amplification due to size-regulated cell growth, in which the cell growth rate roughly scales with the cell size. Such a cell growth regulation has mainly been observed in unconfined single cells[43]. Our results demonstrated a similar mechanism in physiologically confined cells. Lastly, the nucleus and cell size are correlated throughout this two-stage process. Our finding also suggests that the log-normal distribution observed in the steady-state cells can simply arise from morphologically asymmetric divisions. In contrast to AR, where the origin of heterogeneity is mainly attributed to junctional remodeling[17,18], our results indicate that the geometric constraint due to cell-cell interactions mainly determines the final mean size, but not area heterogeneity.

## Actomyosin and osmotic pressure balance regulate NC ratio

Having established that all cells within a monolayer share a similar NC ratio, we next asked what mechanism coordinates the nucleus and cell sizes. Previous studies in single cells have shown that nucleus size can be controlled by either cytoskeleton[44] or intracellular osmotic pressure[45,46]. To test if these mechanisms play a role in propagating the size heterogeneity from cell to nucleus in confluent epithelia, we performed a series of perturbation experiments to investigate the functional requirement of the cytoskeleton, nucleus-cytoskeleton linkage, and osmotic pressure balance in this context (Fig. 3A).

We first investigated whether actomyosin tension regulates nucleus size by measuring the Pearson correlation coefficient of the NC ratio in confluent cells that have been treated with blebbistatin, a non-muscle myosin II inhibitor, or control vehicle (Fig. 3B). We observed a dose-dependent response, as a higher concentration of blebbistatin further attenuates the nucleus-cell area correlation. Notably, although functional actomyosin has been shown to mainly restrict the nucleus height in isolated single cells[47], we here found that it also limits nuclear area and volume in monolayers. To that end, we obtained 3D image stacks of blebbistatin-treated and control nuclei and measured their volumes, since the projected nucleus area change can be due to either volume change or nucleus flattening[47]. We found that the area, volume, and height increased by approximately 40%, 75%, and 25%, respectively (Fig. S8), suggesting a nearly isotropic expansion of the nucleus when myosin activity is inhibited. We also analyzed the changes in cell area and found them to be statistically insignificant (Fig. S9).

We next assessed the contribution from microtubules, another main component of the cytoskeleton, using nocodazole to inhibit microtubule assembly. However, regardless of the concentrations used, inhibition of microtubules alone was insufficient to reduce the nucleus-cell area correlation (Fig. 3C). When mysoin II and microtubules were simultaneously inhibited, we observed a correlation reduction ( ~30% reduction) similar to the blebbistatin-only samples (Fig. 3D, ~20% reduction). These results thus suggest that, in confluent epithelia, actomyosin tension plays a more dominant role than microtubules in coordinating the nucleus size with the cell size.

To further understand how cytoskeleton regulates the NC ratio, we investigated the requirements of the nucleus-cytoskeleton linkage, which transfers the strain from the cytoplasm to the nucleus through the Linker of Nucleoskeleton and Cytoskeleton (LINC) complex[30]. To inhibit the nucleus-cytoskeleton linkage, we disrupted the LINC complex by expressing a dominant negative GFP-KASH2 (DN-KASH) protein[48]. We found that

disruption of the LINC complex significantly reduces the nucleus-cell area correlation (Fig. 3E), suggesting that the linkage between the nucleus and cytoskeleton is required for regulating nucleus size.

Besides the cytoskeleton, osmotic pressure balance between the cytoplasm and the nucleus has also been proposed to be a main regulator of nuclear-to-cytoplasmic volume ratio[45,46]. In this context, the osmotic pressure difference across the nuclear envelope is predominantly driven by active nuclear transport of macromolecules, causing subsequent nuclear size changes. To examine the contribution of osmotic pressure in NC ratio regulation, we first performed a hypo-osmotic shock experiment[46], in which cells were analyzed before and after selective cell membrane permeabilization, followed by an exchange of cell culture medium for a mixture of 95% water and 5% cell culture medium (Fig. 3F). This medium exchange reduced the cytoplasmic osmotic pressure from ~290 mOsm to ~14.5 mOsm (see Materials and Methods), conducive for nuclear expansion. By live imaging cells with nuclear-BFP and plasma membrane-GFP, we found that nuclei significantly swelled after the osmotic pressure drop, while cell size remained roughly constant (Fig. 3G). Quantitative measurement showed that the nucleus size increased by ~50% (Fig. 3H), while the nucleus area and cell area correlation was significantly reduced (Fig. 3I), likely due to the disproportionate increase in nuclear size across the sample. Notably, this increase in nuclear size occurred without a corresponding substantial change in cell area (Fig. S9), suggesting that the increase in the NC ratio is predominantly driven by a significant enlargement of nuclear size. Osmotic pressure is, therefore, critical for maintaining the NC ratio.

Because osmotic pressure is in part controlled by active transportation of macromolecules across the nuclear envelope[49], we next investigated if nuclear transport is required for NC correlation by inhibiting nuclear export using selinexor, a selective exportin-1 inhibitor (Fig. 3J)[50]. Consistent with the hypo-osmotic shock experiment described above, we found that nuclei swelled by ~1.4 fold in area (Fig. 3K, L) 24 h after selinexor treatment with no significant changes in cell area (Fig. S9). The corresponding volume increase was also validated by analyzing 3D image stacks (Fig. S8). This observation of nuclear expansion is anticipated since molecules could not be shuttled out of the nucleus, leading to an increased osmotic pressure within the nucleus. Importantly, we also observed that the selinexor-treated cells exhibited a significantly lower nucleus-cell area correlation (Fig. 3M), suggesting that the macromolecule homeostasis between nuclear and cytoplasmic compartments is required for regulating nucleus size in confluent epithelia. Collectively, our results show for the first time that both osmolarity and the cytoskeleton play an essential role in NC regulation in confluent epithelia. A comprehensive statistical analysis of the six Pearson correlation coefficient measurements reported in Fig. 3B–D was performed in Fig. S10.

## Nucleus size impacts histone modifications

Is the morphological variability of cells merely a by-product of upstream biological events? Alternatively, could the nucleus size heterogeneity have downstream biological impacts? In the last few decades, seminal studies have demonstrated that physical confinement of cells has profound influences on chromatin organizations, inducing changes in epigenetics and gene expression[24,51]. Motivated by these findings, we first examined different histone modifications, which control the physical properties of chromatin and the corresponding epigenetic states. For instance, Histone H3 Lysine 27 trimethylation (H3K27me3) is associated with gene silencing, or repression, and with the formation of facultative heterochromatin, while Histone H3 Lysine 9 acetylation (H3K9ac) is an euchromatic mark associated with gene activation[52]. By immunostaining H3K27me3 in confluent MDCK cells (Fig. 4A) and then analyzing the correlation between nucleus area and H3K27me3 intensity, we found that the H3K27me3 intensity was anti-correlated with the nucleus area (Fig. 4B). To control for the intrinsic dependence of intensity measurement on the nucleus size and potential variations in staining and imaging, we normalized the H3K27me3 intensity to the DAPI intensity[53]. The anti-correlation between the normalized H3K27me3 intensity and nucleus area was further confirmed by analyzing isolated nuclei with imaging flow cytometry (Fig. S11). Our observed anti-

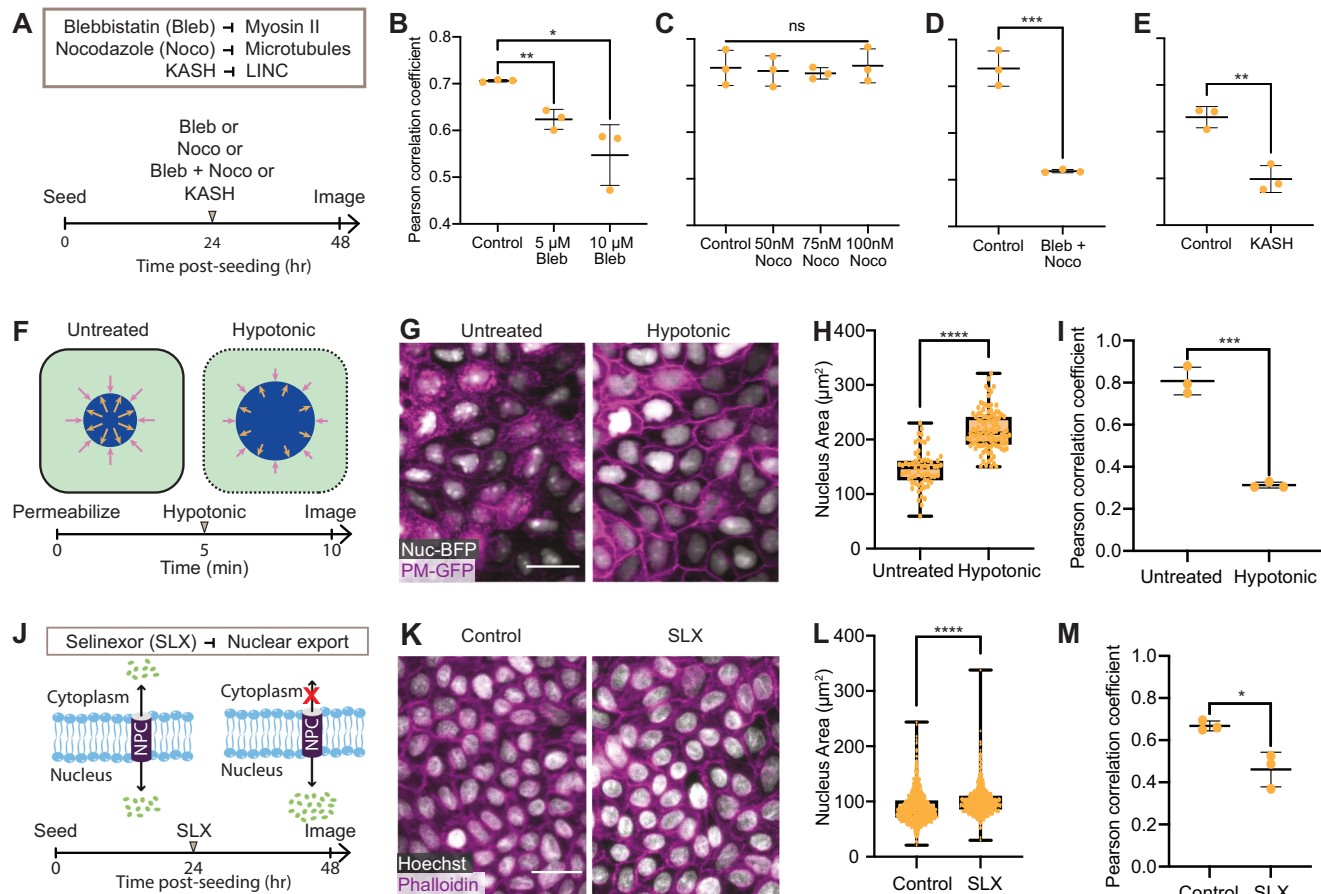

**Fig. 3 | Actomyosin and osmotic pressure balance coordinate the cell and nucleus areas. A** Experimental overview. Inhibitions of myosin II (blebbistain or Bleb), microtubules (nocodazole or Noco), and the nucleo-cytoskeletal coupling through LINC (KASH) were performed for 24 h before sample characterization. Pearson correlation coefficient of cell area and nucleus area in (**B**) Bleb-treated cells, correlation measurements conducted using N = 4751, 9444, and 5248 for control, 5 μM, and 10 μM Bleb-, respectively; (**C**) Noco-treated cells, correlation measurements conducted using N = 3415, 2456, 1999, and 1612 for control, 50 nM, 75 nM, and 100 nM Noco-, respectively; (**D**) Bleb- and Noco-treated cells, correlation measurements conducted using N = 3415 and 1795 for control and Bleb- and nNoco-, respectively; and (**E**) KASH cells, correlation measurements conducted using N = 5376 and 6680 for control and KASH, respectively. **F** Schematic illustrating the hypotonic experiment and timeline. Cells were first treated with digitonin for 5 min to selectively permeabilize the cell membrane (dotted outline). Cell culture medium was subsequently exchanged with a 95% water solution, inducing a decrease

in osmotic pressure. Purple and gold arrows indicate osmotic pressure into and out of the nucleus, respectively. Reduction of arrow size in the hypotonic cell illustrates a decrease in osmotic pressure. **G** Image of control and hypotonic-shocked cells. Nuc-BFP/PM-GFP cells were used for real-time visualization of cell and nucleus morphology. Scale bar = 25 μm. **H** Nucleus area of untreated and hypotonic-shocked cells. **I** Pearson correlation coefficient of cell area and nucleus area of untreated and hypotonic-shocked cells. **J** Schematic illustrating that selinexor (SLX) inhibits nuclear exportation through the nuclear pore complex (NPC), increasing the intranuclear osmotic pressure. Cells were treated with SLX for 24 h before characterization. **K** Image of control and SLX-treated cells. Scale bar = 25 μm. **L** Nucleus area of control and SLX-treated cells. **M** Pearson correlation coefficient of cell area and nucleus area of control and SLX-treated cells. All correlation measurements shown in (**B, C, D, E, I, M**) were conducted using N = 100 cells, in which all p < 0.05. "ns", *, **, ***, ****, refer to p ≥ 0.05, <0.05, <0.01, <0.001, and <0.0001, respectively. 3 biological replicates were used for all analyses.

correlation results illustrate that larger nuclei contain less H3K27me3 compared to smaller nuclei. We repeated the same measurement using the developing E12.5 (Fig. S12) and E12.5 (Fig. 4C) mouse epithelia and observed a similar correlation (Fig. 4D), indicative of a conserved process in cultured cells and in vivo systems.

Next, we assessed the expression of the euchromatic mark H3K9ac in confluent MDCK cells (Fig. 4E). In contrast to the H3K27me3 result, we observed a positive correlation between the nucleus size and H3K9ac intensity, where H3K9ac was upregulated in larger nuclei (Fig. 4F). This result was reproduced in the E12.5 mouse embryonic epithelium (Fig. 4G, H). Our results in both MDCK cells and mouse embryos suggest that there is a universal mechanism dictating H3K27me3 and H3K9ac expression through nucleus size regulation.

To further characterize how nucleus size impacts the chromatin state, we performed systematic spatial analyses of the histone mark intensity within individual nuclei. We focused on two higher-order organization features: chromatin radial distribution and compaction. Here, the radial

distribution reports how chromatin is arranged with respect to the nuclear lamina[54], whereas chromatin compaction impacts DNA replication[55,56] and damage response[57]. To quantify the radial distribution, we defined the outer 20% of a nucleus as the nuclear periphery and the inner 80% as the center (Fig. 4I). We then calculated the intensity ratio between these two nuclear regions. To illustrate the different levels of euchromatin aggregation, we first showed that larger nuclei exhibited more chromatin aggregation when compared to smaller nuclei (Fig. 4J). Such an euchromatin aggregation morphology was then confirmed by calculating the coefficient of variation (CV) of the histone intensity[58].

By characterizing the radial distribution of H3K9ac in crowded MDCK cells, we found that larger nuclei have less centralized H3K9ac, as illustrated by the positive correlation between the periphery-center ratio and nucleus area (Fig. 4K). We also found that this observed correlation does not strongly depend on the split ratio between the nucleus center and the periphery (Fig. S13). This finding confirms that nucleus size can influence chromatin spatial distribution. To confirm our findings on the

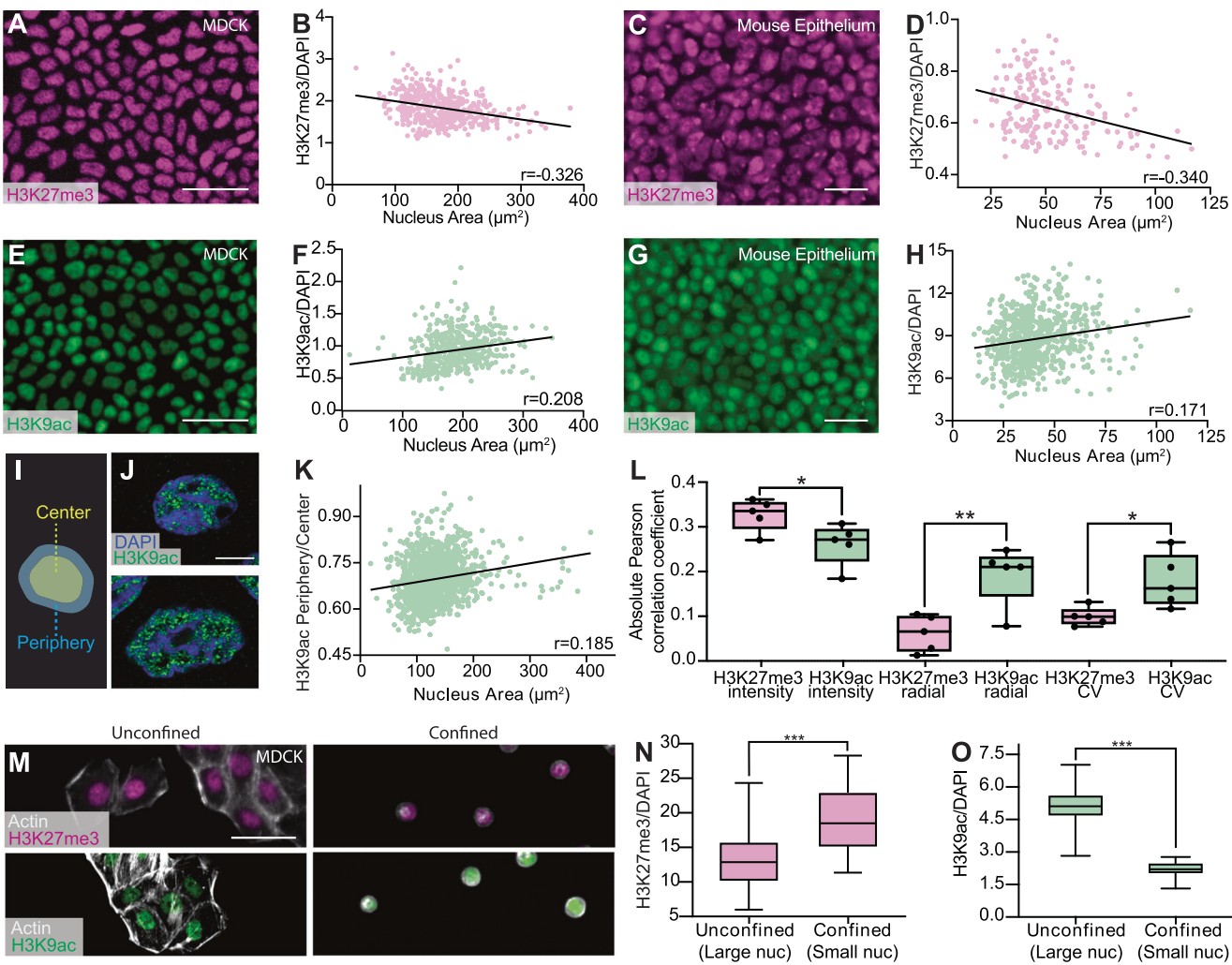

**Fig. 4 | Nucleus size regulates the expression of histone marks. A** Image of MDCK cells stained with Histone H3 Lysine 27 trimethylation (H3K27me3) illustrating its variable expression levels in different cells. Scale bar = 50 μm. **B** Correlation between normalized H3K27me3 expression level and nucleus area in MDCK cells. $N = 461$. 95% confidence interval is $[-0.447, -0.194]$. **C** Image of an E12.5 mouse epithelium stained with H3K27me3. Scale bar = 25 μm. **D** Correlation between normalized H3K27me3 expression and nucleus area in the mouse epithelium. $N = 130$. 95% confidence interval is $[-0.459, -0.209]$. **E** Image of MDCK cells stained with Histone H3 Lysine 9 acetylation (H3K9ac). Scale bar = 50 μm. **F** Correlation between normalized H3K9ac expression and nucleus area in MDCK cells. $N = 1130$. 95% confidence interval is $[0.183, 0.293]$. **G** Image of an E12.5 mouse epithelium stained with H3K9ac. Scale bar = 25 μm. **H** Correlation between normalized H3K9ac expression and nucleus area in the mouse epithelium. $N = 713$. 95% confidence interval is $[0.0988, 0.241]$. **I** Schematic of the nucleus region split for radial distribution analysis. Yellow shaded region occupying 80% of total nucleus area is classified as the center, while the outer 20% shaded in blue is classified as the periphery. **J** Airyscan image of a small (top) and large (bottom) nucleus of crowded MDCK cells illustrating differences in the radial distribution of H3K9ac. Scale bar = 5 μm. **K** Correlation between H3K9ac periphery-center ratio and nucleus area of crowded MDCK cells. $N = 1246$. 95% confidence interval is $[0.131, 0.238]$. **L** Summary of the absolute value of the Pearson correlation coefficient between nucleus area and intensity or spatial distribution analyses in crowded MDCK cells. **M** Images of unconfined (left) and confined (right) MDCK cells stained with H3K27me3 (top) or H3K9ac (bottom). Cells were confined using circular 10 μm fibronectin micro-patterned substrates. Scale bar = 50 μm. **N** Box and whisker chart demonstrating an increase of normalized H3K27me3 intensity in confined cells. Data were obtained from $N = 59$ and 75 nuclei derived from three independent stamp samples for unconfined and confined cells, respectively. **O** Box and whisker chart showing a decrease of normalized H3K9ac intensity in confined cells. $N = 50$ and 52 nuclei derived from three independent stamp samples for unconfined and confined cells, respectively. *, **, and ***, refer to $p < 0.05$, $<0.01$, and $<0.001$ respectively. $N = 3$ biological replicates were used for all plots.

H3K9ac spatial distribution, we performed Airyscan and stimulated emission depletion (STED) microscopy to further visualize stained chromatin marks (Fig. S14). By analyzing 114 representative nuclei, we reproduced the positive correlation between the H3K9ac periphery-center ratio and nucleus area (Pearson correlation coefficient ~0.29, Fig. S14). Lastly, we systematically analyzed how CV and periphery-center ratio correlates with nucleus area for both H3K9ac and H3K27me3, and summarized all the correlation coefficients in Fig. 4L. The summary highlights that nuclear size mainly affects the organization of euchromatin, labeled by H3K9ac, more than heterochromatin, labeled by H3K27me3.

Our observed effect of nuclear size on chromatin states is reminiscent of previous micro-patterning constraint studies. In our system, the physical confinement arises from the neighboring cells. To examine if geometric constraint alone without the effect of cell-cell adhesion can induce a similar histone modification change, we micro-patterned 10 μm-diameter fibronectin disks on an anti-adherent substrate to mimic the geometric constraint found in crowded MDCK cells (Fig. 4M). We showed that the microprinted fibronectin was able to confine individual cells. In comparison, subconfluent, unconfined cells exhibited a larger area and irregular cell shape (Fig. 4M). Consistent with both previous literature and our findings in monolayers, we found that confinement upregulated H3K27me3 (Fig. 4N)

and downregulated H3K9ac (Fig. 4O)[20,59]. Furthermore, we observed that confined cells displayed reduced cell and nuclear areas (Fig. S9), aligning with our finding that smaller nuclei are associated with increased H3K27me3 levels. Our finding confirms that geometric constraint can play a role in regulating chromatin modifications. Compared to previous confinement studies, which typically impose a constraint much smaller than the nucleus size[59], our finding suggests that a relatively mild, physiologically relevant emerging during crowding is sufficient to alter chromatin states.

## Nuclear area as a primary morphological predictor of H3K27me3 levels

Expanding on the identified association between nuclear sizes and histone mark levels, we conducted a systematic linear multivariable analysis to evaluate the significance of this nuclear property. This analysis included seven morphological features of MDCK cells and nuclei, as well as two textural properties (Fig. 5A). We first validated this approach by confirming the strong 1-to-1 relationship between cell and nucleus areas, establishing area as a key feature linking cells and nuclei (Fig. 5B). Specifically, a single-variable regression, using cell area as the sole predictor, showed a strong correlation between predicted and measured nucleus area, consistent with previous findings above. Incorporating additional cellular morphologies into a multivariable model did not enhance the predictive accuracy compared to cell area alone, even when employing a nonlinear Gaussian process regression (GPR) model. Also, canonical correlation analysis (CCA), which identifies and quantifies linear relationships between nuclear and cellular features by maximizing correlations between their canonical variates, only slightly improved the nucleus morphology prediction. These results confirm that the nucleus-cell area correlation is primarily pairwise, with other morphological features playing a minimal role.

We then sought to identify the primary descriptors of phenotypic heterogeneity in our MDCK cell population using principal component analysis (PCA). The biplot shown in Fig. 5C revealed a relatively isotropic distribution along the top three principal components, each contributing similarly to the total variance. H3K27me3 levels, cell/nucleus area, and cell aspect ratio were approximately orthogonal, indicating that heterogeneity cannot be explained by a single morphological variable and that predicting H3K27me3 levels requires integrating multiple morphological features. This finding further suggested that the H3K27me3 regulatory pathways may involve multiple cellular properties with nonlinear interactions, consistent with the observed moderate correlation between H3K27me3 levels and nuclear size.

We then investigated whether additional morphological features could improve H3K27me3 level prediction (Fig. 5D). Notably, incorporating more nuclear or cellular morphological features did not significantly enhance prediction accuracy, confirming nucleus area as the primary morphological predictor of H3K27me3 levels. However, prediction accuracy improved with the inclusion of NC ratios, and further enhancement was achieved using a nonlinear GPR model. The high predictive power is demonstrated by the strong correlation ($r = 0.720$) between predicted and measured H3K27me3 levels (Fig. 5E). To evaluate robustness, we reconstructed the spatial distribution of H3K27me3 by regenerating the cell and nuclear layouts of an MDCK monolayer, with each nucleus color-coded based on normalized H3K27me3 levels (maximum = 1, minimum = 0) (Fig. 5F).

The inclusion of the NC ratio and consideration of nonlinear relationships among predictors likely improves predictive accuracy by unifying various aspects of nuclear and cellular morphology and function that collectively influence H3K27me3 levels, effectively capturing the interplay between nuclear and cytoplasmic factors regulating histone modifications. Our findings suggest that the relationship between nuclear size, cell size, and histone modifications is nonlinear, with nuclear area serving as the primary morphological predictor of H3K27me3 levels.

Building on these findings, we investigated how H3K27me3 levels are associated with cellular properties when the nucleus-cytoskeleton linkage is disrupted. Analysis of DN-KASH cells revealed that, after LINC disruption, nucleus area can no longer effectively predict H3K27me3 levels (Fig. 5G). However, incorporating all morphological features, NC ratios (Fig. 5H), and nonlinear relationships (Fig. 5I) restored prediction accuracy to a level comparable to that in the control samples. This restored predictive power suggests that LINC disruption reprograms the relationship between histone mark levels and morphological properties. To evaluate this reprogramming, we performed a dropout analysis, systematically removing individual morphological features from the linear regression model and assessing the impact on prediction accuracy. Feature importance rankings revealed distinct trends: in control samples, nucleus area, nucleus shape index, and NC aspect ratio were key predictors, whereas in DN-KASH samples, cell perimeter and circularity emerged as dominant predictors (Fig. 5J). These results suggest that LINC disruption does not abolish the association between H3K27me3 levels and cellular morphology but instead reprograms it.

## Nucleus size regulates H3K27me3 levels via UTX

Because our multivariable analysis revealed that nucleus size is the most dominant morphological predictor of H3K27me3, we next set out to understand how nucleus size alters chromatin states. To do this, we investigated the levels of histone-modifying enzymes, which are responsible for post-translational modifications of histones[60]. We first characterized the dependence of histone demethylase levels on nucleus size by immunostaining MDCK cells for the lysine-specific demethylase UTX (Fig. 6A) and compared the normalized UTX/DAPI intensities in the top and bottom 20% of nuclei based on size (Fig. 6B). This analysis indicates that UTX levels are correlated with nucleus size (Fig. S15), consistent with our previously observed anti-correlation between the H3K27me3 level and nucleus size. We also assessed how the lysine methyltransferase EZH2 levels relate to nucleus size (Fig. 6C), but did not observe a significant correlation (Fig. 6D). Since UTX and EZH2 play antagonistic roles, we next calculated their intensity ratio (Fig. 6E), which could determine the overall H3K27 methylation state. To that end, we observed a lower EZH2/UTX intensity ratio in larger nuclei, corroborating the notion that UTX facilitates the coordination between nucleus size and H3K27me3 levels.

To further examine the functional roles of these enzymes, we administered GSK-J1 and DS3201 to inhibit UTX and EZH2, respectively (Fig. 6F). We validated both drugs' effects by observing increased H3K27me3 expression upon GSK-J1 treatment and reduced expression upon DS3201 treatment (Fig. 6G). By calculating the Pearson correlation coefficient between nucleus size and H3K27me3/DAPI level, we found that the anti-correlation observed in the control group was abolished by either treatment (Fig. 6H), demonstrating that both enzymes are required to regulate H3K27me3 levels in response to changes in nucleus sizes. Lastly, since we found that cytoskeletal tension regulates the NC ratio and it has been suggested to affect the histone-modifying enzyme recruitment[20], we also examined how it affects histone modifications in our system. We found that blebbistatin significantly reduced the UTX level (Fig. 6I, J). Importantly, we observed reduced correlation between nucleus area and H3K27me3 (Fig. 6K) or H3K9ac (Fig. S16) levels, indicating that actomyosin is critical for coordinating chromatin modifications with nuclear sizes. We also investigated the potential impact of histone mark levels on nuclear heterogeneity by analyzing nuclear size distributions. The results revealed similar probability density functions across control, DS3201-treated, and GSK-J1-treated cells in both subconfluent and crowded conditions (Fig. S17).

To elucidate how nuclear size regulates H3K27me3 through UTX, we conducted confinement experiments using micro-patterning to restrict cell spreading and assessed normalized UTX levels at 3, 5, and 8 hours post-seeding (Fig. 6L). Quantitative analysis revealed that UTX accumulated in the nuclei of control, unrestrained cells, whereas nuclear UTX levels were significantly reduced in confined cells (Fig. 6M). Using hypotonic perturbations, we validated this trend by observing increased UTX levels in nuclei enlarged by hypo-osmotic shock (Fig. S18). In the

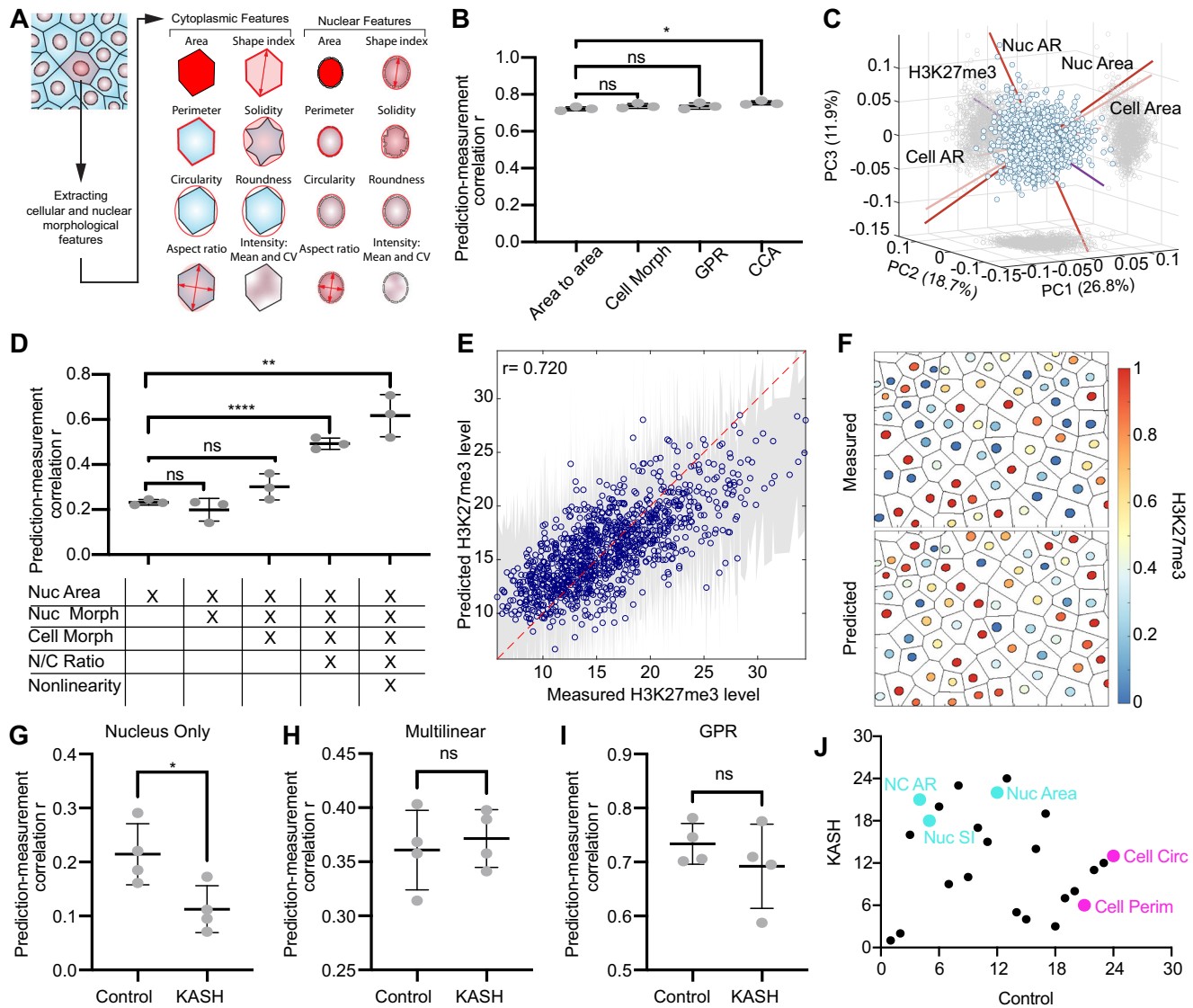

**Fig. 5 | Nucleus-cell coordination is primarily pairwise and nucleus area is a key mophological predictor for H3K27me3. A** Schematic illustrating cytoplasmic and nuclear features used in morphological analyses. CV refers to coefficient of variation of DAPI intensity. **B** Prediction-measurement correlation with nucleus area. Axis range has been optimized to highlight differences between groups. Group titled "Cell Morph" refers to cell morphology illustrated in (**A**). GPR and CCA denote Gaussian process regression and canonical correlation analysis, respectively. **C** Principal component analysis biplot of cell morphology, nuclear morphology, and H3K27me3 levels. PC indicates principal component. **D** Prediction-measurement correlation for H3K27me3 levels. Groups titled "Cell Morph" and "Nuc Morph" refer to morphologies illustrated in (**A**). Nonlinearity group utilizes GPR model. **E** H3K27me3 level predicted using GPR versus measured level. Gray shaded band represents the GPR 95% confidence interval. Red dashed line denotes a perfect correlation ($r = 1$). $N = 1080$. **F** Illustration of measured H3K27me3 levels in MDCK cells (top) and predicted levels using GPR (bottom). Color bar indicates normalized H3K27me3 level. **G** H3K27me3 prediction-measurement correlation for control and DN-KASH (KASH) using nucleus area as the sole predictor. **H** H3K27me3 prediction-measurement correlation obtained using multi-linear regression. **I** H3K27me3 prediction-measurement obtained using GPR. **J** Predictor importance rank comparison scatter plot between control and KASH cells. Key morphological features for control and KASH samples are emphasized in fuchsia and cyan, respectively. Correlation measurements were conducted using $N = 376, N = 335$, and $N = 446$ cells for (**B, D**); $N = 1277, N = 1346$, and $N = 1101$ cells for the control condition in (**G–I**); and $N = 1262, N = 1438$, and $N = 1147$ cells for the KASH condition in (**G–I**). "ns", *, **, ***, ****, refer to $p \geq 0.05$, $< 0.05$, $< 0.01$, $< 0.001$, and $< 0.0001$, respectively. 3 biological replicates were used for (**B–F**), while 4 biological replicates were used for (**G–J**).

micro-patterning experiments, we further observed a corresponding increase in H3K27me3 levels (Fig. 6N). These findings suggest that nuclear size reduction (Fig. S9) hinders nuclear UTX accumulation, leading to an increase in H3K27me3 levels, a process that takes approximately 8 h to become evident.

## Discussion
Integrating all findings from this study, we propose a model (Fig. 6P) in which uneven cell division generates cell size variability, which is transmitted to nuclear size variability through actomyosin tension and

intracellular osmotic pressure balance. This nuclear size variability, in turn, drives variation in histone modifications, partly by modulating UTX expression.

Our paralleled statistical analyses of both the cell morphology and nucleus morphology have provided a detailed statistical description of morphological variabilities in epithelial cells. These analyses unveiled a constant correlation between the cell and the nucleus size, such that they are co-scaled as cell density increases. These results have two key implications for the epithelial cell jamming transition, when the cell collective experiences crowding and shifts from a fluid-like state, allowing for cell rearrangement,

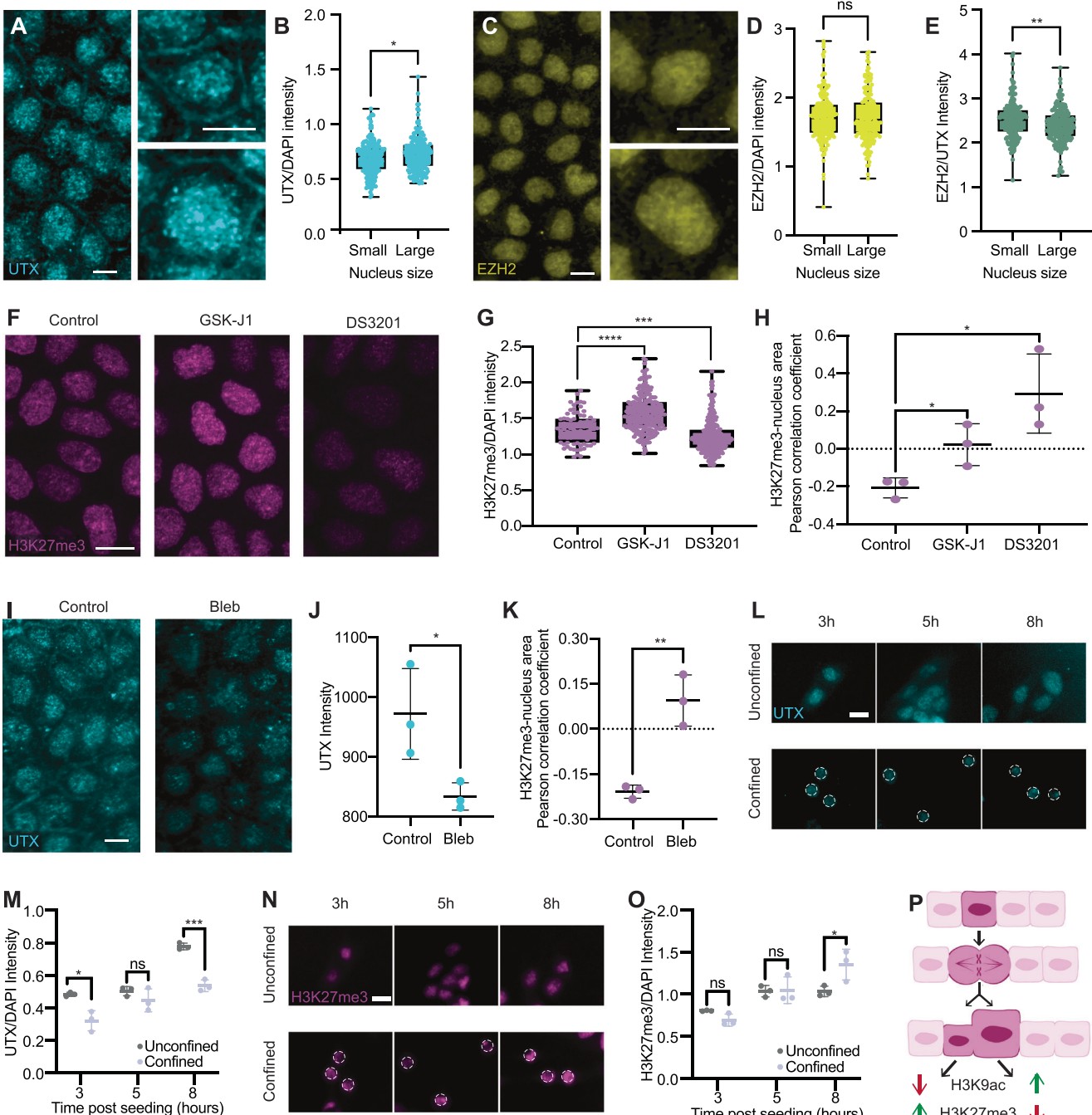

**Fig. 6 | UTX coordinates with nucleus size and regulates H3K27me3 levels.**
**A** MDCK cells stained for UTX with example images of small (upper right) and large nuclei (bottom right). **B** The UTX/DAPI intensity of the smallest 20% of nuclei (Small) is significantly lower than that of largest 20% of nuclei (Large). N = 295 pooled from 3 biological replicates. **C** MDCK cells stained for EZH2 with example images of small (upper right) and large nuclei (bottom right). **D** EZH2 levels displayed no significant difference between small and large nuclei. N = 295 pooled from 3 biological replicates. **E** EZH2/UTX intensity ratio is higher in small nuclei than in large nuclei. N = 295 pooled from 3 replicates. **F** MDCK cells stained for H3K27me3 in control, GSK-J1 and DS3201 samples. **G** GSK-J1 and DS3201 respectively increased and decreased the H3K27me3 levels. N = 77, 217, and 294 for control, GSK-J1, and DS3201, respectively. **H** Pearson correlation coefficient between nucleus size and H3K27me3 intensity for control, GSK-J1, and DS3201 treated cells. GSK-J1 and DS3201 treatments reduced the anti-correlation between nucleus size and H3K27me3 intensity. Dotted line denotes no correlation. **I** MDCK cells stained for UTX in control and blebbistatin-treated (Bleb) samples. **J** Bleb samples exhibited

lower UTX levels than control. N = 6. **K** Pearson correlation coefficient between nucleus size and H3K27me3 intensity for control and Bleb samples. Dotted line denotes no correlation. **L** Fluorescent images illustrating UTX levels in unconfined (top) and 10 μm-confined (bottom) cells. White dashed outline in confined cells denotes nuclear contour. **M** Quantification of UTX intensity normalized to DAPI intensity in unconfined and confined cells. **N** Same as (**L**) but for H3K27me3. **O** Same as (**M**) but for H3K27me3. **P** Proposed mechanism of how heterogeneous cell morphology generates diversity in nucleus size and chromatin states. In a crowded epithelial monolayer, a mother cell divides unevenly into two daughter cells with different sizes, which persist thereafter. Each cell size then propagates through actomyosin tension and intracellular osmotic pressure balance to determine the corresponding nucleus size. Cell size heterogeneity thus gives rise to nucleus size heterogeneity, which in turn contributes to the varying UTX levels and chromatin modifications. Scale bar = 10 μm for (**A, C, F, I**). Scale bar = 20 μm for (**L, N**). p < 0.05 for (**B, D, F, H, K**). "ns", *, **, ***, ****, refer to p ≥ 0.05, <0.05, <0.01, <0.001, and <0.0001, respectively. 3 biological replicates were used for all analyses.

https://doi.org/10.1038/s42003-025-07677-w                                                                                                    **Article**

to a solid-like state, typically characterized by reduced cell motility[61,62]. First, we demonstrated that the nearly universal distribution of cell size and AR can be extended to the nucleus morphology. This nucleus-cell correlation shows that cells and their nuclei share a similar morphological signature during jamming transition. Therefore, as the cells undergo jamming transition that is facilitated by direct intercellular interactions, nuclei are concurrently influenced by this jamming process through the cytoskeletal tension and osmotic pressure balance. Second, our findings suggest that the variabilities of cell AR and area are governed by distinct mechanisms. Previous studies have demonstrated that cell AR variations are largely impacted by remodeling the intercellular junction[63], which is a multi-cellular effect due to packing. Our result shows that cell area differences, in contrast, are predominantly determined at cell birth, which is mainly a unicellular effect.

Our finding that variability in cell size can arise from uneven cell divisions unveils a source of cell size variability and its universality in a biological system[64,65]. The current understanding of variability origin is mainly built on conjectures predicting PDFs that are subsequently tested by experiments with a focus on the role of packing geometry. Given that cell divisions can generate daughter cells with different sizes[66], our findings here provide a framework on how cell divisions can introduce cell and nucleus size variations within a population, that are subsequently maintained by each cell to generate the observed morphological heterogeneities inherent in the system. Furthermore, our observed persisting cell size disparity supports a previously proposed cell growth model by which the molecular synthesis, self-assembly, and transport determine the cell growth rate[67]. In future studies, it would be important to determine how cell-cell forces regulate cell cycles and growths, thereby determining the final size of steady-state cells.

To understand how the nucleus-cell size coordination is achieved, we investigated the roles of both active nuclear exportation of macromolecules and actomyosin functions. Our results obtained using a confluent cell layer highlight the importance of intracellular osmotic pressure balance in nuclear size regulation, as well as its requirement for coordinating the nucleus-cell size ratio. Similarly, perturbations of actomyosin functions or the physical connection of cytoskeleton to the nucleus via LINC resulted in loss of nucleus-cell size coordination as well. Therefore, both osmotic pressure balance and actomyosin are critical for coordinating cell and nucleus sizes. However, it remains unclear whether inhibition of actomyosin hinders nucleus-cell size regulation mechanically or via nuclear transport[68]. Future experiments will decipher the two possibilities by observing the elastic response of nucleus upon laser ablation of cytoskeleton. Finally, because asymmetric actomyosin localized at cell cortex can generate unequal forces during cell division to produce daughter cells of unequal sizes[64,65], it will be important in the future to determine actomyosin's role in driving size heterogeneities, in addition to the nucleus-cell size correlation we have identified here.

How nucleus morphology can control gene expression, such as through chromatin modifications, is an active research topic[49]. Leveraging the cell-to-cell variability in our steady-state monolayer system, we directly tested the regulation of chromatin by nucleus sizes without perturbing transient intracellular molecular events, and observed a positive or negative correlation between the nucleus area and the H3K9ac or H3K27me3, respectively. When compared to previous micro-patterning experiments that examined isolated single cells, our observation unveils the pivotal role of nucleus geometry in regulating chromatin organizations in an epithelial monolayer setting. Our result also unveils that UTX facilitates these nucleus size-driven chromatin modifications, in which the UTX levels are controlled by the nucleus size and actomyosin tension. Our finding of the correlation between UTX levels and nucleus size is consistent with recent experiments, where nuclear envelope curvature was found to regulate the nuclear pore complex conformation, altering molecular weight-dependent nucleocytoplasmic transport[68]. This finding is also consistent with recent theoretical predictions, which posit that the nucleus volume alone can impact the chromatin state by altering the intra-nuclear electrostatic potential[69] and

surface area to volume ratio, affecting expression of genes associated with the nuclear lamina[70]. Lastly, previous studies have shown that chromatin state and its interaction with the nuclear envelope can determine nuclear morphology[71]. Our work highlights that a reverse regulation can too take place, as the nuclear morphology, controlled by the cytoskeleton and nuclear transport, can modulate the levels and distributions of different chromatin marks.

Finally, our findings here provide a potential mechanism by which cells in living tissues can generate diversity in cell shapes, fates, and behaviors. For instance, concomitant changes in cell/nucleus sizes and differentiation can occur following asymmetric cell divisions during *Drosophila* neuroblast differentiation[72], mouse keratinocyte differentiation[73], and mouse blastocyst patterning[74]. In a pathological setting, cancer cells with larger nuclei are often linked to a more metastatic state[75]. It is therefore plausible that variations in cell and nucleus sizes, regardless of how they are produced, help generate a spectrum of chromatin modifications that in turn modulate gene expression and fate changes across different cells. Taken together, our results unveil the process through which cell and nucleus sizes are coordinated to control chromatin modifications and distribution, thus translating variations in cell sizes into differences in chromatin organizations. Cell morphological heterogeneity that is present in any tissue may therefore play an important role in generating cell diversity.

## Methods
### Cell culture and drug treatment
All experiments conducted using Madin Darby Canine Kidney cells (MDCK II cell line) were cultured in MEM-$\alpha$ (Fisher Scientific, 12561-056) supplemented with 10% fetal bovine serum (FBS) (Fisher Scientific, 12662-029) and 1% Penicillin-Streptomycin (Fisher Scientific, 15140-122). The MDCK II cell line was a gift from Jeffrey Fredberg at Harvard University. HaCaT cells (AddexBio, T0020001), were cultured under low-calcium conditions for growth and propagation and under high-calcium conditions for differentiation, as outlined in previous HaCaT culturing protocols[76]. DMEM was used as the base medium for both low-calcium and high-calcium media. Low calcium media was supplemented with 2% 200 nM L-glutamine, 1% of 3.0 mM calcium chloride solution, and 10% low calcium FBS. High-calcium media was made by adding 20 mL of 200 nM L-glutamine, 10 ml of 280 mM calcium chloride solution, and 100 ml of low-calcium FBS to 780 mL of DMEM. Low-calcium FBS was prepared by adding 0.38 g of Chelex 100 resin to 50 mL of FBS and incubating for one hour at 4 °C on a tube rotator. MDCK and HaCaT cells were maintained at 37 °C and 5% $CO_2$ with humidity. For all MDCK and HaCaT assays, cells were passaged when they reached ~80% confluence using Trypsin/EDTA solution (Fisher Scientific, 25300-054). Cells utilized in time course experiments and subconfluent experiments were seeded at 30,000 cells/cm², while cells utilized in confluent experiments were seeded at 80,000 cells/cm².

To inhibit non-muscle Myosin II, microtubules, and nuclear export, blebbistatin (Millipore Sigma, B0560), nocodazole (Millipore Sigma, M1404), and selinexor (Selleckchem, KPT-330) were used, respectively. Concentrations used were 5 $\mu$M and 10 $\mu$M for blebbistatin, 50 nM, 75 nM, and 100 nM for nocodazole, and 10 $\mu$M for selinexor. When combined, 10 $\mu$M blebbistain was mixed with 30 nM nocodazole. GSK-J1 (Selleckchem, S751) and DS3201 (Selleckchem, S8926) were respectively used as a histone demethylase inhibitor and a histone methylase inhibitor at a concentration of 5 $\mu$M. These optimal concentrations were determined by titrating doses to achieve their primary functions without causing other adverse effects, such as apoptosis. All pharmacological agents were administered for 24 h at 37 °C and 5% $CO_2$ once cells reached ~90% confluence.

### Mouse line and procedure
All animal procedures were conducted in compliance with the animal protocols approved by the UCLA Institutional Animal Care and Use Committee (Protocol Number ARC-2019-013). We have complied with all relevant ethical regulations for animal use. Embryos were not sexed and were selected at random for all experiments. No wild-type animals were

used; the only inclusion criterion was Cre-induced membrane-GFP positivity in epithelial cells.

$K14^{Cre77}$ and $R26^{mT/mG78}$ mice were group housed and genotyped as previously published. To generate $K14^{Cre}$; $R26^{mT/mG}$ embryos for experiments, timed pregnancy was set up between $K14^{Cre}$; $R26^{mT/mG}$ mice, and E12.5 embryos were harvested subsequently. Pregnant mice were euthanized by $CO_2$ followed by cervical dislocation. Both membrane GFP-positive male and female embryos were selected at random and used in all experiments. Embryonic tissues were harvested and fixed in 4% paraformaldehyde (PFA) in PBS overnight at 4 °C. Tissues were subsequently washed with PBS three times and stained as previously described. All mice were maintained in the University of California Los Angeles (UCLA) pathogen-free animal facility. All mouse experiments were approved by the UCLA Institutional Animal Care and Use Committee (Protocol Number ARC-2019-013). No design study protocol was prepared before this study.

### Immunostaining

For cell culture experiments using HaCaT and MDCK cells, cells plated in a 4-chamber Ibidi slide (Ibidi, 80426) were fixed in 10% neutral buffered formalin with 0.03% Eosin (Sigma-Aldrich, F5304-4L) for 10 minutes at room temperature. Cells were then simultaneously permeabilized and blocked using a mixture of 2% Donkey Serum (Fisher Scientific, D9663-10ML) with 0.25% Triton X-100 diluted in PBS with calcium and magnesium (Fisher Scientific, 14040-133) at room temperature for 30 min. After washing three times with PBS, cells were then incubated in a primary antibody solution for 30 min at room temperature. The primary antibodies utilized were H3K27me3 (Cell Signaling Technology, 9733S, dilution 1:800), H3K9ac (Active Motif, 61663, concentration 2 µg/ml), EZH2 (Cell Signaling Technology, 3147S, dilution 1:200), and UTX (Cell Signaling Technology, 33510S, dilution 1:200). The cells were then washed three times using PBS before incubating with the secondary staining solution using anti-mouse Alexa Fluor 488 and anti-rabbit Alexa Fluor 647 (Invitrogen, A21042 and 4414S, respectively; concentration 4 µg/ml) secondary antibodies for 30 min at room temperature. Nuclei were labeled using DAPI (Invitrogen, D1306, concentration 1 µg/ml) was added along with the secondary antibody.

For mouse embryo staining, the same protocol was utilized except with incubation times of 1 hour for the permeablization/blocking buffer, 24 hours at 4 °C for the primary staining solution, 3 hours for the secondary staining solution, and 2 hours for the PBS washes.

### Imaging and analysis

Fluorescent images were acquired using either a confocal microscope (RCM1 with Nikon Eclipse Ti-E, NIS-Elements software) or a widefield fluorescent microscope (Etaluma LS720 Lumaview 720/600-Series software). A 20 × WI/0.95 NA, 60 × WI/1.00 NA, or 20 × /0.75 NA objective was used for images taken using confocal microscopy. A 20×/0.40 NA objective was used for the widefield fluorescent images. The imaging conditions were consistently maintained across all experiments. To quantify fluorescent intensity, z-projected images obtained from z-stack imaging using a step size of ~2 µm were used. In time lapse imaging, images were taken using the Etaluma microscope every 5 minutes for 6 days with media changes performed every other day. Morphological segmentation was performed using Cellpose[79] or Trackmate[80]. Maximum intensity projections of z-stacks were utilized for segmentation. Quantifications were performed using MATLAB (version R2023a) or ImageJ (version 2.1.0/1.53c). For analyses investigating the spatial localization of histone modifications, fluorescence intensities were normalized to the area of the center and periphery regions. In all relevant analyses, all cells at the edge of the image were excluded since their corresponding morphology could not be accurately measured. Additionally, actively dividing cells were excluded for analysis, as they exhibit non-representative morphology and DAPI intensity. Confounders were not controlled.

For the multivariable analyses shown in Fig. 5, all computations were performed using the MATLAB Regression Learner App and custom scripts.

Morphological features were quantified with ImageJ measurement tools. The quantified morphological features are defined as follows: shape index is the ratio of the perimeter to the square root of the area, solidity is the ratio of the area to the convex area, circularity is the ratio of the area to the square of the perimeter, roundness is the ratio of the area to the square of the major axis length, aspect ratio is the ratio of the major axis length to the minor axis length, and the coefficient of variation is the ratio of the standard deviation to the mean. Regression prediction accuracy was evaluated using the Pearson correlation coefficient between predicted (MATLAB Regression Learner App) and ground truth values. Nonlinear regression modeling employed Gaussian Process Regression (GPR) with a squared exponential kernel, chosen for its ability to handle noise, sparsity, and biological variability while providing probabilistic predictions with uncertainty quantification. For H3K27me3 level analysis, DAPI mean intensity, standard deviation, and coefficient of variation were excluded from the predictors list, as their direct correlation with normalized H3K27me3 levels (H3K27me3/DAPI) lacks biological significance. Predictor importance was assessed via single-variable dropout analysis, where each predictor was excluded individually, and the regression model was recalculated. The predictors were ranked based on their importance, quantified by the reduction in the Pearson correlation between predicted and measured values caused by their dropout.

### Hypotonic shock experiment

MDCK cells with nuclear BFP and plasma membrane GFP were cultured in a 4-chamber Ibidi slide (Ibidi, 80426) until confluent. Z-stack images of both BFP and GFP channels were then acquired using a 60× WI objective. Following this, $10^{-4}$% V/V digitonin was added to the cell culture media for 5 minutes and the same imaging protocol was repeated. To introduce a hypotonic environment, the cell culture media containing digitonin is then replaced with a solution 95% Milli-Q water and 5% MEM-α, in which the corresponding osmolarity is then calculated based on the concentration of individual solutes in the final solution. After a 4 minute incubation with the hypotonic solution, cells were imaged using the same imaging procedure described above.

### MDCK cell transduction

To label plasma membrane with GFP and nucleus with BFP for live imaging, cells were first transfected with a pAcGFP1-Mem vector to produce a plasma-membrane line. These cells were seeded at a density of $2 \times 10^4$ cells per well in 12-well plates. Three days after seeding, cells were infected with 250 µL of lentivirus expressing nucleus BFP and 8 ng/µL polybrene (MilliporeSigma, TR-1003-G). The plate was immediately centrifuged at $500 \times g$ for 1 h at room temperature following the infection. After centrifugation, 750 µL culture medium was added without removal of lentiviruses. BFP/GFP double positive clonal populations were sorted by FACS using a BD FACS Aria H. To generate the KASH reporter, MDCK cells were seeded at a density of $2 \times 10^4$ cells per well in 12-well plates. Cells were transfected three days after seeding using Lipofectamine Reagents (Invitrogen, 18324012) with 0.75 µg GFP-KASH2 vector (Addgene plasmid no. 187017) and 0.25 µg PiggyBac transposon vector. After neomycin selection, clonal populations were sorted by FACS using a BD FACS Aria H.

### Micro-patterning

Micro-patterned polydimethylsiloxane (PDMS) stamps, containing circular pillar arrays, were obtained from Research Micro Stamps (Clemson, SC). PDMS stamps underwent sonication in an ethanol bath for sterilization. Following sterilization, the pillars of the stamp were incubated with 20 µg/mL fibronectin for 30 min at room temperature. The PDMS stamp was then affixed to hydrophobic, untreated petri dishes to transfer the fibronectin. The surface was then treated with Anti-Adherence Rinsing Solution (STEMCELL technologies, 07010) for 45 min to minimize nonspecific cell binding.

## Statistics and reproducibility

GraphPad PRISM (version 10) and MATLAB (version R2023a) were used to perform statistical analysis and create figures. Data is presented as mean ± standard deviation. For correlative analyses of all presented scatter plots, both the Pearson correlation coefficient and the Spearman correlation coefficient were calculated in addition to the $p$ value, false discovery rate (FDR), and confidence interval (CI), all of which are summarized in Table S1. FDR was evaluated using a Monte Carlo simulation that randomly permutes the $XY$ values of data points. $P > 0.05$ are denoted as not significant (ns), while $p$-values $\leq 0.05$, $\leq 0.01$, $\leq 0.001$, and $\leq 0.0001$, are represented as *, **, ***, and ****, respectively. Statistical comparisons were calculated using unpaired two-sided $t$ tests. The data points presented in Fig. 4L were pooled from three independent biological replicates and randomly resampled. No bootstrapping methods were applied in any analyses. Pearson correlation coefficient was calculated by dividing the covariance of two variables by the product of their standard deviations. The Spearman correlation coefficient was obtained by calculating the Pearson correlation coefficient between the ranks of two variables. For box and whisker plots, the box represents the interquartile range (IQR), whiskers denote the minimum and maximum points, and the line represents the median of the dataset. For the dot plots, the error bars denote the standard deviation. The sample sizes, number of replicates, and descriptions of replicates are detailed in the corresponding figure captions. For animal experiments, no a priori calculation was used. Sample size was determined using standard lab practice of $n \geq 3$. As these are correlation studies of different cellular features using control animals, no randomization was carried to allocate samples into different experimental group.

## Data availability

All data supporting the findings presented in this manuscript are provided in the main text and Supplementary Information. Additional data are available from the corresponding author upon reasonable request. Source data are included in Supplementary Data 1, while data used for the multivariable analysis are provided in Supplementary Data 2.

## Code availability

All algorithms used to analyze the data in this manuscript are described in detail in the Methods section. Cell segmentation was conducted using Cellpose 2.0. All analyses in this manuscript were performed using MATLAB (R2021a) and Excel (2021), utilizing built-in functions with default parameters. Custom MATLAB code is available from the corresponding author upon reasonable request[81]. https://doi.org/10.5281/zenodo.14795857.

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

## Acknowledgements
The authors thank Ming-Heng Tsai and Sanako Takano for assisting with the cell transduction and mouse sample preparation, respectively. The authors also acknowledge the kind help from Samanta Negrete Muñoz, Jerry Chen, Iris Sloan, Alyssa Gee, and Simone Horowitz for processing the data and images. The authors are grateful to Gerard Wong, Samuel Safran, Dan Deviri, Bill Gelbart, and Alexander Hoffmann for insightful scientific discussions. A.B. and N.Y.C.L. were supported by NSF (CBET-2244760, CMMI-2029454, DBI-2325121) and NIH NIGMS (R35GM146735). J.K.H. was supported by NIH NIDCR (R01DE030471). D.B. was supported by the NSF (DMR-2046683), the NIH NIGMS (R35GM15049), the NSF (PHY-2019745) and the Alfred P. Sloan Foundation.

## Author contributions
A.B., J.K.H., and N.Y.C.L conceived and designed the study. A.B., Z.D.L., and A.J.M. performed the experiments and were aware of group allocations at all stages of the experiment. All authors analyzed and interpreted the data. A.B., Z.D.L., D.B., J.K.H., and N.Y.C.L. wrote and edited the manuscript. All authors read and approved the manuscript.

## Competing interests
The authors declare no competing interests.
