## [Transparent Peer Review file · Communications Biology]

Regulation of Chromatin Modifications through Coordination of Nucleus Size and Epithelial Cell Morphology Heterogeneity

Corresponding Author: Professor Neil Lin

Version 0:

Reviewer comments:

Reviewer #1

(Remarks to the Author)

In the manuscript "Regulation of Chromatin Modifications through Coordination of Nucleus Size and Epithelial Cell Morphology Heterogeneity" by Bermudez et al, the authors study how nuclear and cell size changes during epithelial crowding. The authors propose that size heterogeneity arising from asymmetric cell division results in heterogeneity in histone modifications. Using small molecule inhibitors, they show that nuclear size is regulated by actomyosin contractility, osmotic pressure, and nuclear export. They identify the histone demethylase UTX as a key player in the size-dependent changes in histone methylation.

Overall, the figures are clear and the manuscript is well-written. It was a joy to read and I commend the authors for that. More details on the Methods, especially the quantification of the immunofluorescence (IF) images would be helpful for the reader. My main feedback is that I don't yet see how the cause-effect relationship between the histone acetylation/methylation levels and nuclear shape can be deduced from the experiments. I highlight my comments below and invite the authors to revise their manuscript in response.

Major comments

1. It is not clear why the heterogeneity in histone marks should be a consequence of nuclear heterogeneity rather than a cause. If the authors want to test this, they will have to use the UTX and EZH2 inhibitors in the sparse conditions and show that nuclear heterogeneity is identical to the control conditions. Here, all we can say is that there is a correlation between nuclear size heterogeneity and chromatin epigenetic heterogeneity.
2. Please clarify where the nuclei and cells were segmented from single z-planes vs maximum intensity projections of z-stacks
3. Do the authors remove the outliers from the scatter plots in any way? In some of the figures, the correlation appears to arise from a handful of outliers, which, if removed, would result in no correlation (e.g. Figs. 4K, all panels of S11)
4. Confidence intervals on the Pearson correlation: The authors should report a confidence interval for the Pearson correlation. Some of the correlation coefficients are rather small which the authors report as "significant". This would be misleading.
5. Controls in Fig 3 look very different: A follow-up to #3 above is that the Pearson correlation coeffs look very different for the conditions shown in Fig 3 B-D. What is the confidence interval for each of the correlation measurements?
6. STED images: The authors present a correlation coefficient from 16 points in Fig S12. As the authors mention in other figures, they calculate the correlation from $> N=100$ cells. This correlation measurement from such a small number of cells is not meaningful. The authors should collect a similar number of cells if they want to report these data. The confidence interval of the correlation coefficient should be reported as well.
7. In each figure legend, please indicate how many cells were imaged for each panel (not merely $> n$ cells) and how many independent biological replicates were collected

8. Figure 1: Please report the correlation coefficients in the figure/legend as well

9. Which statistical tests were used? The methods do not mention how statistical comparisons were made. Were the samples bootstrapped?

10. Show complete distributions: There are several instances of boxplots in the manuscript. Please replace those to show the entire distribution (the probability density function) for conditions that are being compared. This will make it easier to gauge heterogeneity in the data.

11. Center vs periphery fluorescence intensity measurements: Are the fluorescence intensities normalized to the area of the center and periphery regions? If not, please do that before reporting the results. This will be a consistent measure of the density of histone marks.

12. The authors do not present the center vs periphery results for H3K27me3 in the main figure 4. There is no panel for H3K27me3 with the 80-20% periphery-center split. Why is that?

13. How does the nucleus area change in the actomyosin/osmotic pressure/nuclear export experiments? Is the NC ratio changing mainly because of nuclear shape change?

14. Similarly, there is no quantification of nuclear/cell area for the conditions presented for the micropatterning experiments

15. Fig 4A argues that smaller nuclei have higher H3K27me3 intensity. The opposite is shown for Fig 4N. The same is true for the H3K9ac data in 4O vs 4F, H. The main difference between these two experiments is that while the nuclei shape is modulated the same way, the cells in the micropatterning experiments do not have cell-cell interactions. Doesn't this argue that cell-cell signaling is important and that geometry is not the only determinant?

16. S13 A shows that UTX levels are correlated with nuclear area not anti-correlated as suggested in the text.

Minor comments:

1. Not clear how this statement follows from the co-scaling of nuclear and cell size: "These results have two important implications on the epithelial cell jamming transition, in which a tissue transitions from a fluid-like state that permits cell rearrangement to a solid-like state that is characterized by tight cell packing, increased cell-cell adhesion, and reduced cell motility"
2. Blebbistatin is inactivated by blue light used for GFP excitation (PMID 15641783). Are these data from live or fixed cells?
3. Fig 3, why don't all the cells have actin in the phalloidin staining?
4. S8 the colors are difficult to see. Please change this to grayscale
5. Change modulation of UTX expression to be down/up regulated

Reviewer #2

(Remarks to the Author)

Authors discussed the progression of epithelial cell morphology and nuclear morphology and its effect on chromatin modification. While authors results provide a hint of morphological heterogeneity to determining chromatin states, it largely lacks the mechanistic explanation regarding the alteration of cell/nuclear morphology and its consequence of epigenetic modification. They might consider following issues prior to publication of their article.

For instance, does the prestress existing in the epithelia have any impact on the chromatin state? Time-dependent changes of nucleus and cell morphology and/or NC ratio in response to the external physical stimuli would also lead any changes of gradual and/or acute reorganization of chromatin structure. Comments on these queries would greatly improve the quality of the paper.

Version 1:

Reviewer comments:

Reviewer #1

(Remarks to the Author)

In the revised manuscript, the authors have done well to address the comments raised by myself and Reviewer 2.

I have only two further comments before publication:

1. The new multivariate analysis is very interesting, but I could not find sufficient details on the various parameters that the authors calculate. For example, what is the difference between circularity and roundedness? I request the authors to provide further details on the analytical parameters in the Methods section for the reference of the readers.
2. This is a very rich dataset, so in the spirit of open science, I request the authors to publish all their measurements along with the paper so that readers can download the datasets and either benchmark their own data against these, or draw further inferences from all the cell morphology parameters.

Minor: A couple of instances of 'respectfully' should be corrected to 'respectively'

Dear Editor,

We are pleased to resubmit our revised manuscript titled “*Regulation of Chromatin Modifications through Coordination of Nucleus Size and Epithelial Cell Morphology Heterogeneity.*” We sincerely thank you and the reviewers for your thoughtful and constructive feedback, which has significantly strengthened our work. We were particularly encouraged by Reviewer 1’s positive comments that our manuscript was “a joy to read” and by their commendation of the clarity of our figures.

The reviewers identified two key areas for improvement:

1. To establish stronger causality between histone methylation levels and nucleus size.
2. To provide a more detailed mechanistic understanding of how morphological alterations impact histone mark expression levels.

In response, we conducted additional experiments and expanded our analyses to address these concerns comprehensively. The revisions include substantial new data and refined interpretations that enhance the rigor and depth of our original findings:

- **Addressing causality between histone marks and nucleus size:**
 - We performed two complementary experiments to establish causality. In the first, we reduced nucleus size using microprinting-based confinement and demonstrated that this manipulation resulted in downstream changes in histone mark levels.
 - In the second experiment, we treated cells with DS3201 (a UTX inhibitor) or GSK-J1 (an EZH2 inhibitor) and observed no significant changes in nucleus size across both sparse and crowded conditions. These findings provide compelling evidence that histone mark heterogeneity arises as a consequence of nuclear size heterogeneity.
- **Providing mechanistic insights into morphology and histone mark levels:**
 - As suggested by Reviewer 2, we conducted additional microprinting experiments to investigate UTX transport and its role in regulating H3K27me3 levels. We found that reductions in nucleus size led to a decrease in nuclear UTX levels and a corresponding increase in H3K27me3 levels—a process that unfolds over approximately 8 hours.
- **Advancing system-level understanding:**
 - We performed a multivariable analysis of key morphological features of both cells and nuclei, which revealed a nonlinear relationship between nuclear size, cell size, and histone modifications. Notably, nuclear area was confirmed as the primary morphological predictor of H3K27me3 levels, thus both reinforcing and refining our original conclusion.

These new findings, alongside other minor revisions and clarifications detailed in the accompanying point-by-point response, significantly enhance the robustness and depth of our reported mechanisms. We believe these improvements address the Reviewers’ concerns and elevate the manuscript’s contribution to the field. In summary, this study is among the first to link chromatin state variation to asymmetric cell division, by identifying how a cell regulates its nucleus size and histone-modifying enzyme levels.

Thank you for your consideration of our revised submission. We look forward to your feedback and are happy to provide additional clarifications if needed.

Sincerely,
Jimmy Hu and Neil Y.C. Lin

Reviewer #1 (Remarks to the Author):

In the manuscript “Regulation of Chromatin Modifications through Coordination of Nucleus Size and Epithelial Cell Morphology Heterogeneity” by Bermudez et al, the authors study how nuclear and cell size changes during epithelial crowding. The authors propose that size heterogeneity arising from asymmetric cell division results in heterogeneity in histone modifications. Using small molecule inhibitors, they show that nuclear size is regulated by actomyosin contractility, osmotic pressure, and nuclear export. They identify the histone demethylase UTX as a key player in the size-dependent changes in histone methylation.

Overall, the figures are clear and the manuscript is well-written. It was a joy to read and I commend the authors for that. More details on the Methods, especially the quantification of the immunofluorescence (IF) images would be helpful for the reader. My main feedback is that I don't yet see how the cause-effect relationship between the histone acetylation/methylation levels and nuclear shape can be deduced from the experiments. I highlight my comments below and invite the authors to revise their manuscript in response.

Major comments

1. It is not clear why the heterogeneity in histone marks should be a consequence of nuclear heterogeneity rather than a cause. If the authors want to test this, they will have to use the UTX and EZH2 inhibitors in the sparse conditions and show that nuclear heterogeneity is identical to the control conditions. Here, all we can say is that there is a correlation between nuclear size heterogeneity and chromatin epigenetic heterogeneity.

Response: We thank the reviewer for suggesting that the causality of our finding can be further validated by performing additional experiments. Following the Reviewer's suggestion, we have conducted a new experiment by treating the cells with DS3201 (a UTX inhibitor) or GSK-J1 (an EZH2 inhibitor) and measuring the nucleus size for both sparse and fully crowded samples. As shown in the figure below, we found that the nucleus size was not significantly impacted by the applied inhibitions (middle and right columns), where the nucleus size distributions universally collapsed for both sparse and crowded systems (C & D). This unchanged nucleus size provides evidence indicating that the histone mark heterogeneity is a consequence of nuclear heterogeneity, rather than being the cause.

To further demonstrate that the nuclear size regulates histone mark levels, we conducted an additional experiment, in which we utilized microprinting to reduce nucleus size and quantify corresponding changes in H3K27me3 levels. We found that confinement leads to reduced nuclear UTX localization, leading to an increase in H3K27me3 expression. This result confirms that heterogeneity in histone marks is a consequence of nuclear size heterogeneity. Additional details on this experiment are presented in our response to Reviewer 2 as well as in Figs. 6 L-O in the manuscript. Corresponding discussion is presented in the “Nucleus size regulates H3K27me3 levels via UTX” section in the manuscript.

Revisions: The nucleus area heterogeneity results have been added to Fig. S17, along with the corresponding discussion in the “Nucleus size impacts histone modifications” section of the Results: “... coordinating chromatin modifications with nuclear sizes. We further demonstrated that histone mark heterogeneity arises from nuclear heterogeneity by examining how DS3201 and GSK-J1 treatments impact nuclear size. We found that the nucleus size distributions across control, DS3201, and GSK-J1 universally collapsed in both subconfluent and crowded samples (Fig. S17), suggesting that H3K27me3 levels do not significantly regulate nuclear size.”

2. Please clarify where the nuclei and cells were segmented from single z-planes vs maximum intensity projections of z-stacks

Response: We appreciate the Reviewer for emphasizing the importance of this information. For all morphological analyses in the paper, segmentation was performed on maximum intensity projections of z-stacks to ensure accurate measurement of the entire area of both cells and nuclei.

Revisions: We have updated the manuscript to include this information in the “Imaging and Analysis section” of the “Materials and Methods” section: “... performed using Cellpose (79) or Trackmate (80). Maximum intensity projections of z-stacks were utilized for segmentation.”

3. Do the authors remove the outliers from the scatter plots in any way? In some of the figures, the

correlation appears to arise from a handful of outliers, which, if removed, would result in no correlation (e.g. Figs. 4K, all panels of S11)

Response: For all datasets reported in this study, we carefully examined potential outliers to validate each data point. The “outliers” identified in the figures referenced by the Reviewer did not present any justifiable reason for exclusion, even if their values were considerably above or below the mean. To assess whether the reported correlation was influenced by these “outliers,” we performed the suggested analysis. Specifically, we removed both the top and bottom 5% of data points that did not fall within the clustered region of each scatter plot, and recalculated the Pearson correlation coefficients. We focused on Figs. 4K, S11F, and S11H, as their initial Pearson correlation coefficients were found to be >0.16 and statistically significant. For Figs. 4K and S13F, our exclusion criteria removed cells with nuclei sizes smaller than $60 \mu\text{m}^2$ or larger than $215 \mu\text{m}^2$, and for Fig. S13H, nuclei sizes smaller than $105 \mu\text{m}^2$ or larger than $275 \mu\text{m}^2$ were excluded. After removing these data points, we found that the Pearson correlation coefficient increased for all analyses, with p-values <0.0001 , as shown in the figures below (top left: Fig. 4K, top right: Fig. S13F, bottom middle: Fig. S13H). This analysis indicates that our observed correlation did not arise from the outliers in the scatter plots.

4. Confidence intervals on the Pearson correlation: The authors should report a confidence interval for the Pearson correlation. Some of the correlation coefficients are rather small which the authors report as “significant”. This would be misleading.

Response: We appreciate the Reviewer for emphasizing the importance of the confidence interval measurement. This analysis has already been performed for all datasets and was summarized in Table S1 of the previous version of the manuscript. Table S1 provides comprehensive significance assessments, including p-values, 95% confidence intervals, and false discovery rates (FDR).

Revisions: To make this information easily accessible to readers in the revised manuscript, we have further included the confidence intervals for all main figures in their respective figure captions.

5. Controls in Fig 3 look very different: A follow-up to #3 above is that the Pearson correlation coeffs look very different for the conditions shown in Fig 3 B-D. What is the confidence interval for each of the correlation measurements?

Response: We appreciate the Reviewer’s suggestion to examine the consistency of the Pearson correlation coefficient measurements in our inhibition experiments (Fig. 3). To address this, we performed a statistical analysis on the Pearson correlation coefficients obtained from six biological replicates across two independent control experiments (shown in Figs. 3B and 3C/D). The control condition for the DN-KASH experiment was excluded from this analysis, as the control cell line had been transduced with a Dox-inducible DN-KASH construct but was not treated with doxycycline.

As depicted in the figure below, the 6 Pearson correlation coefficients exhibit similar values, with variations generally falling within the confidence intervals indicated by the error bars. To assess statistical significance, we calculated the Pearson correlation coefficients’ magnitude (r) and sample size (n) and applied Fisher’s Z-transformation followed by a Z-test. This approach allowed us to compare correlations between datasets (replicates), with Z-test values greater than 1.96 or less than -1.96 indicating significant differences. The Z-test values were then converted to p-values using a two-tailed probability based on the standard normal distribution. The summarized results are presented in the tables below.

Our analysis identified a significant difference only in the final replicate, compared to the others, while the remaining datasets showed no significant differences. This suggests that, overall, the Pearson correlation coefficients across the six biological replicates from the two experimental runs are consistent. Furthermore, to assess batch-to-batch variation in the Pearson correlation coefficient measurements, we calculated the p-value between the two independent control experiments. The analysis revealed no significant difference between the two experiments, as shown in the final figure below.

We have also revisited the control sample image in Fig. 3K by re-performing the experiment with optimized drug concentrations and treatment durations to minimize non-specific effects on nuclear and cytoskeletal remodeling. The figure has been updated accordingly with the new data, and further details about this update are provided in our response to Comment 13.

Z-test values	3b-Rep 1	3b-Rep 2	3b-Rep 3	3c/d-Rep 1	3c/d-Rep 2	3c/d-Rep 3
3b-Rep 1		-0.093	0.194	-1.497	0.268	-4.191
3b-Rep 2			0.275	-1.379	0.345	-3.992

3b-Rep 3				-1.588	0.075	-4.131
3c/d-Rep 1					1.620	-2.304
3c/d-Rep 2						-4.098
3c/d-Rep 3						

p-values	3b-Rep 1	3b-Rep 2	3b-Rep 3	3c/d-Rep 1	3c/d-Rep 2	3c/d-Rep 3
3b-Rep 1		0.92608	0.84598	0.13439	0.78853	0.00003
3b-Rep 2			0.78332	0.16792	0.73038	0.00007
3b-Rep 3				0.11239	0.93986	0.00004
3c/d-Rep 1					0.10533	0.02124
3c/d-Rep 2						0.00004
3c/d-Rep 3						

Revisions: We have included the corresponding figure and tables in the Supplementary Information as Fig. S10 and added a sentence to the main manuscript: “A comprehensive statistical analysis of the six Pearson correlation coefficient measurements reported in Figs. 3B-D was performed in Fig. S10.”

6. STED images: The authors present a correlation coefficient from 16 points in Fig S12. As the authors mention in other figures, they calculate the correlation from $> N=100$ cells. This correlation measurement from such a small number of cells is not meaningful. The authors should collect a similar number of cells if they want to report these data. The confidence interval of the correlation coefficient should be reported as well.

Response: We agree that the histone mark spatial distribution results would be more statistically robust with an analysis of over 100 nuclei. To address this, we conducted additional imaging and analyses of 114 nuclei using Airyscan semi-superresolution imaging. The results are presented below. With the inclusion of these additional data points, the correlation has increased to $r = 0.4733$, with a p-value < 0.0001 and a confidence interval of $[0.3169 \text{ to } 0.6046]$.

Revisions: The updated figure, along with the Pearson correlation coefficient, p-value, and confidence interval, have been included in Fig. S14.

7. In each figure legend, please indicate how many cells were imaged for each panel (not merely $> n$ cells) and how many independent biological replicates were collected

Response: We agree that specifying the number of cells imaged and the number of independent biological replicates is important.

Revisions: As such, we have updated the manuscript to include this information in the figure legends for each relevant figure.

8. Figure 1: Please report the correlation coefficients in the figure/legend as well

Revisions: We have updated Figures 1D and 1H to include the Pearson correlation coefficients in the main figure, and these values have been added to the figure caption as well.

9. Which statistical tests were used? The methods do not mention how statistical comparisons were made. Were the samples bootstrapped?

Response: We thank the reviewer for identifying the need to further clarify the statistical comparison method. In our analyses, statistical comparisons were performed using unpaired two-sided t-tests, and no bootstrapping was applied.

Revisions: To clarify this, we have revised the sentence, “P-values were calculated using unpaired two-sided t-tests,” to “Statistical comparisons were calculated using unpaired two-sided t-tests” in the “Data Analysis and Statistics” section of the “Materials and Methods.” Additionally, we have added the following sentence to confirm that no bootstrapping was performed: “No bootstrapping was applied in any analysis.”

10. Show complete distributions: There are several instances of boxplots in the manuscript. Please replace those to show the entire distribution (the probability density function) for conditions that are being compared. This will make it easier to gauge heterogeneity in the data.

Response: We thank the reviewer for the valuable suggestion to show the complete distributions of the data. We have carefully reviewed all the box and whisker plots presented in the main figures and supplementary information (Figs. 1B-C, 1F-G, 3B-E, 3H-I, 3L-M, 4L, 4N-O, 6B, 6D-E, 6G-H, 6J-K, S5B, S8B-D, and S14B). Upon review, we found that these plots either already include corresponding scatter plots or are primarily intended to convey the Pearson correlation coefficient for different biological replicates, or to highlight global changes between conditions, rather than to specifically characterize sample heterogeneity.

However, we recognize the value of displaying the full distributions where relevant. To address this, we are happy to provide the probability density functions for any figures where the reviewer believes sample heterogeneity needs to be better emphasized. Please let us know if there are specific figures in which you would like to see this additional information.

11. Center vs periphery fluorescence intensity measurements: Are the fluorescence intensities normalized to the area of the center and periphery regions? If not, please do that before reporting the results. This will be a consistent measure of the density of histone marks.

Response: In our previous version, we normalized the mean intensity to the corresponding regional area to consistently measure the density of histone marks.

Revisions: More detailed information on this analysis method has been added to the “Data Analysis and Statistics” section of the “Materials and Methods”: “For analyses investigating the spatial localization of histone modifications, fluorescence intensities were normalized to the area.”

12. The authors do not present the center vs periphery results for H3K27me3 in the main figure 4. There is no panel for H3K27me3 with the 80-20% periphery-center split. Why is that?

Response: We thank the reviewer for indicating that it would be useful to provide more comprehensive measurements regarding the H3K27me3 spatial localization analysis.

Revisions: We have included the suggested analysis of the 80-20% periphery-center split for H3K27me3, which is consistent with the results of all other reported H3K27me3 spatial localization analyses. Since the results were not significant, they were not initially included as a main figure. The results are as shown in the figures below. These additional analyses have been added to Fig. S13.

13. How does the nucleus area change in the actomyosin/osmotic pressure/nuclear export experiments? Is the NC ratio changing mainly because of nuclear shape change?

Response: We thank the reviewer for requesting clarification regarding the change in the NC ratio. As demonstrated in the figures below, we observed that the cell area remains consistent between control and treated conditions across all actomyosin inhibition and hypotonic shock experiments. In contrast, nuclear size showed a statistically significant increase. Therefore, the observed change in the NC ratio is primarily attributed to alterations in nuclear size. Regarding the nuclear export inhibition experiment, upon further examination, we found that the inhibition slightly reduced cell density, impacting the cell area. We have therefore optimized the drug treatment (final concentration = 5 µM, treatment time = 12 hours) to minimize the drug’s impact on cell density. In the updated dataset, we confirmed no change in cell area, with a significantly increased nucleus area.

Revisions: We have added the figures below as Fig. S9. Also, the following sentences have been added to the “Actomyosin and osmotic pressure balance regulate NC ratio” section to provide discussion on the NC ratio change in these experiments: “Additionally, we found that the increase in the NC ratio is primarily

driven by a significant increase in nuclear size, with no change in cell area across the various treatments (Fig. S9)”

14. Similarly, there is no quantification of nuclear/cell area for the conditions presented for the micropatterning experiments

Response: As suggested by the reviewer, we conducted additional analyses quantifying cell and nucleus area for both confined and unconfined cells. As illustrated by the plots below, the results show that confinement significantly reduces both cell and nucleus area. Given that H3K27me3 levels are increased in confined cells, its downregulation is likely linked to the smaller nuclear area.

Revisions: Both plots shown below have been included in Fig. S9, and we have added the following sentence to the “Nucleus size impacts histone modifications” section of the Results: “... Furthermore, we found that confined cells exhibit reduced cell and nucleus area (Fig. S9), with the decrease in nuclear size potentially contributing to the increased H3K27me3 levels observed in these cells.”

15. Fig 4A argues that smaller nuclei have higher H3K27me3 intensity. The opposite is shown for Fig 4N. The same is true for the H3K9ac data in 4O vs 4F, H. The main difference between these two experiments is that while the nuclei shape is modulated the same way, the cells in the micropatterning experiments do not have cell-cell interactions. Doesn't this argue that cell-cell signaling is important and that geometry is not the only determinant?

Response: We apologize for the presentation of the data in Figs. 4N and 4O, which may appear visually inconsistent with their respective counterparts in Figs. 4B and 4F. To clarify, Figure 4N shows that confined (smaller) nuclei exhibit higher H3K27me3 intensity, consistent with the findings in Figs. 4A and 4B. Similarly, Fig. 4O demonstrates that confined (smaller) nuclei exhibit lower H3K9ac intensity, aligning with the results presented in Figs. 4C and 4D. The data in Figs. 4N and 4O were ordered to present the control (unconfined) cells before the experimental (confined) group. Together, these results highlight that cell geometry, rather than cell-cell signaling, is the primary factor influencing H3K9ac and H3K27me3 expression.

Revisions: To prevent confusion, we have explicitly labeled the box and whisker plots in Figs. 4N and 4O to clearly indicate which condition corresponds to small and large cells, ensuring a more intuitive interpretation of the results, consistent with the data shown in Figs. 4B, D, F, and H.

16. S13 A shows that UTX levels are correlated with nuclear area not anti-correlated as suggested in the text.

Response: We thank the reviewer for pointing out this typo.

Revisions: We have corrected the text in the “Nucleus Size Regulates H3K27me3 Levels via UTX” subsection to read: “...bottom 20% of nuclei based on size (Fig. 6B). This analysis indicates that UTX levels are correlated with nucleus size (Fig. S15)...”

Minor comments:

1. Not clear how this statement follows from the co-scaling of nuclear and cell size: “These results have two important implications on the epithelial cell jamming transition, in which a tissue transitions from a fluid-like state that permits cell rearrangement to a solid-like state that is characterized by tight cell packing, increased cell-cell adhesion, and reduced cell motility”

Response: This sentence is not a statement but primarily provides a definition of the cell jamming transition. The two implications are discussed in the subsequent sentences.

Revisions: To better align this sentence with the context of the study, we have revised it as follows: “These results have two key implications for the epithelial cell jamming transition, where the cell collective experiences crowding and shifts from a fluid-like state, allowing for cell rearrangement, to a solid-like state, typically characterized by reduced cell motility.”

2. Blebbistatin is inactivated by blue light used for GFP excitation (PMID 15641783). Are these data from live or fixed cells?

Response: All data for blebbistatin-treated cells were collected from fixed samples to ensure that the effects of treatment remained intact and were not affected by blue light exposure.

3. Fig 3, why don't all the cells have actin in the phalloidin staining?

Response: We agree that the brightness and contrast of the phalloidin channel in Fig. 3K could be improved.

Revisions: We have updated the image with a revised ROI that more clearly displays actin.

4. S8 the colors are difficult to see. Please change this to grayscale

Revisions: As suggested by the Reviewer, we have converted the images in Fig. S8 to grayscale to enhance the visualization of nuclear morphology.

5. Change modulation of UTX expression to be down/up regulated

Revisions: All instances previously referred to as “UTX modulation” have been replaced with the appropriate regulation descriptor. The following sentences were modified accordingly.

“Moreover, we provide insights into the impact of nucleus morphology on chromatin dynamics... heterochromatic mark H3K27me3 through down-regulation of histone demethylase UTX expression.”

“Importantly, cell and nucleus size variations are established and maintained upon each cell doubling event, ... such that nuclear constraint leads to down-regulated UTX expression and thus increased H3K27me3.”

Reviewer #2 (Remarks to the Author):

Authors discussed the progression of epithelial cell morphology and nuclear morphology and its effect on chromatin modification. While authors results provide a hint of morphological heterogeneity to determining chromatin states, it largely lacks the mechanistic explanation regarding the alteration of cell/nuclear morphology and its consequence of epigenetic modification. They might consider following issues prior to publication of their article. For instance, does the prestress existing in the epithelia have any impact on the chromatin state? Time-dependent changes of nucleus and cell morphology and/or NC ratio in response to the external physical stimuli would also lead any changes of gradual and/or acute reorganization of chromatin structure. Comments on these queries would greatly improve the quality of the paper.

We thank the reviewer for their insightful comments regarding the depth of explanation in our study. Before we discuss the additional experiments added to the revised manuscript, we would like to first highlight that the primary contribution of this work is our identification of a biophysical mechanism regulating nuclear size and its influence on histone mark levels, which was uncovered through an innovative correlation analysis leveraging the intrinsic phenotypic heterogeneity within cell monolayers. These promising results established a foundation for investigating how morphological heterogeneity influences chromatin states through underlying molecular mechanisms. Although investigating the molecular mechanism lies beyond the scope of this study, the revised manuscript includes new findings that elucidate how nucleus size modulates UTX levels, which subsequently impact H3K27me3 levels.

Specifically, we conducted three additional experiments: (1) a time-course micropatterning confinement experiment to test how reducing cell/nucleus size temporally affects UTX levels, (2) a hypotonic shock experiment to study how enlarging nucleus size affects UTX levels, and (3) a multivariable analysis to investigate the interplay between cellular and nuclear morphological features on H3K27me3 regulation. These additional investigations collectively reinforce the hypothesis that nuclear size modulates the nuclear transport of UTX, which in turn regulates H3K27me3 levels. In the following sections, we provide a detailed description of each experiment and its primary findings, which elucidate the mechanistic relationship between nuclear size and chromatin state.

Time-course micro-patterning experiment

In response to the Reviewer's suggestion for time-course experiments, we focused on micro-patterning, as it is a well-characterized and relatively non-invasive method for inducing cell and nuclear confinement. Using this approach, we confined MDCK cells following the previously described protocol (10 μm confinement achieved through micro-patterning fibronectin on an anti-cell-adherent substrate). After seeding, cells were allowed to attach and spread for 3, 5, or 8 hours. At each time point, as well as for unconfined control samples, cells were fixed and stained with DAPI, UTX, and H3K27me3. Confocal z-stack imaging was performed, followed by nuclear segmentation using Cellpose and subsequent analyses to quantify normalized levels of UTX and H3K27me3.

Approximately 8 hours after confinement, we observed that confined cells exhibits lower levels of UTX but higher amounts of H3K27me3 in the nucleus when compared to unconfined control cells. This finding indicates that micro-patterning-induced nuclear volume reduction (Figure S9) inhibits proper UTX localization within the nucleus, thereby allowing for the H3K27me3 accumulation, which requires approximately 8 hours to manifest. In contrast, unconfined cells have large nuclei and the corresponding higher nuclear localization of UTX, which counteracts the formation of H3K27me3. To elucidate the mechanisms by which nuclear size reduction decreases UTX levels—such as enhanced export, increased degradation, or diminished import and retention—further experiments are required. These could include live imaging of UTX dynamics or targeted inhibition of nuclear import and export pathways. We plan to perform these experiments in the future.

The above observations thus align with our correlation analysis between H3K27me3 levels and nuclear area in cell monolayers, as shown in Figs. 6L-O. A detailed description of these findings has been included in the results section titled “Nucleus Size Regulates H3K27me3 Levels via UTX.” The added paragraph reads:

“...To elucidate how nuclear size regulates H3K27me3 through UTX, we conducted confinement experiments using micro-patterning to restrict cell spreading and assessed normalized UTX levels at 3, 5, and 8 hours post-seeding (Fig. 6L). Quantitative analysis revealed that UTX accumulated in the nuclei of control, unrestrained cells, whereas nuclear UTX levels were significantly reduced in confined cells (Fig. 6M). Using hypotonic perturbations, we validated this trend by observing increased UTX levels in nuclei enlarged by hypo-osmotic shock (Fig. S18). In the micro-patterning experiments, we further observed a corresponding increase in H3K27me3 levels (Fig. 6N). These findings suggest that nuclear size reduction (Fig. S9) hinders nuclear UTX accumulation, leading to an increase in H3K27me3 levels, a process that takes approximately 8 hours to become evident.”

Hypotonic shock experiment

While the micro-patterning experiments test the effects of reduced nuclear size, hypotonic shock enables us to investigate the impact of nuclear enlargement on UTX levels. Using our previously described protocol, cells were first pretreated with $10^{-4}\%$ v/v digitonin for 5 minutes at 37°C . For one sample, the digitonin was subsequently washed out and replaced with a hypotonic solution composed of 95% water and 5% culture media, which was applied for 10 minutes at 37°C . As a control, another sample was fixed immediately following the 5-minute digitonin treatment without exposure to the hypotonic solution. After the 10-minute hypotonic shock, the treated cells were fixed. Both the control and hypotonic-treated cells were then stained for UTX and H3K27me3. Nuclear segmentation was performed using Cellpose, followed by quantification of UTX and H3K27me3 levels normalized to DAPI.

The results of the hypotonic shock experiment further corroborate our correlation and microprinting findings, demonstrating that nuclear enlargement leads to approximately 30% higher levels of UTX. However, no significant difference in H3K27me3 levels was observed between untreated and hypotonic shock-treated samples. This outcome may be explained by several factors, including non-specific nuclear recruitment of methylases and demethylases or the extended time scale required for H3K27me3 demethylation, as suggested by the 8-hour timeframe observed in the micro-patterning experiment. To

differentiate between these possibilities, in the future, we plan to conduct experiments involving EZH2 inhibition or disruption of nuclear import/export processes. Such experiments fall outside the primary scope of the current study and will not alter our conclusions here.

It is important to note that extending the duration of the hypotonic shock experiment to 8 hours to detect significant changes in H3K27me3 levels is not feasible. Prolonged exposure to the hypotonic solution (>20 minutes) resulted in cell death and detachment of the monolayer.

These findings have been added as Fig. S18.

Multivariable analysis

Our study focuses on harnessing phenotypic diversity, encompassing both cellular morphology and chromatin modifications, to explore their interplay through correlation-based analyses. While we have identified roles for histone-modifying enzymes in regulating nuclear H3K27me3 levels, the primary contribution of this work lies in providing comprehensive analyses for understanding the interdependence between cell morphology, nuclear morphology, and chromatin modifications. To further emphasize this key contribution, we have incorporated a new figure (Fig. 5) showcasing the results of multivariable analysis.

Our multivariable analysis evaluated seven morphological and two textural properties of MDCK cells and nuclei, providing a systematic approach to understanding phenotypic heterogeneity within the population. To identify the primary contributors to this heterogeneity, we first conducted a principal component analysis (PCA). The biplot revealed a relatively isotropic spread along the top three principal components, each accounting for a similar proportion of the total variance. This result highlights the complex nature of phenotypic diversity, which cannot be reduced to a single morphological variable. Importantly, H3K27me3 contributes distinct information that is independent of nucleus and cell area, as evidenced by the orthogonality of these axes.

Cell-nucleus coordination. We revisited cell-nucleus coordination using multivariable analysis, focusing on how cellular features predict nuclear area. This approach was motivated by our earlier findings that nuclear area is a key feature coordinated by cellular properties and influences histone mark levels. First, we validated the strong correlation between the nuclear area predicted by cell area and the measured values, consistent with prior observations. Expanding the analysis to a multivariable model incorporating all nine cellular features did not enhance predictive power for nuclear area, indicating that cell area alone remains the most prominent predictor. Furthermore, the prediction accuracy was sustained when employing a nonlinear regression model (Gaussian Process Regression (GPR)). Finally, we applied Canonical Correlation Analysis (CCA) to identify and quantify linear relationships between nuclear and cellular features by maximizing correlations between their canonical variates. This analysis revealed improved predictive accuracy for nuclear features, highlighting the role of complex relationships in cell-

nucleus coordination. Collectively, our findings suggest that while cell-nucleus coordination involves multifaceted interactions, the correlation between nuclear and cell areas represents the most significant link.

Histone modifications. Our observation of a moderate correlation between H3K27me3 levels and nuclear size suggests that additional factors beyond nuclear size may contribute to the regulation of H3K27me3 during cell crowding. To explore this possibility, we incorporated all nuclear and cellular morphological features into the analysis but did not observe a significant improvement in the accuracy of H3K27me3 level predictions. This finding indicates that nuclear area is the primary predictor among the tested features. However, the prediction accuracy improved when the nuclear-to-cytoplasmic (NC) ratio was included and was further enhanced by employing a nonlinear Gaussian Process Regression (GPR) model, highlighting the potential importance of the NC ratio and nonlinear relationships in regulating H3K27me3 levels. The improved prediction accuracy suggests that the balance between nuclear and cellular morphology, beyond just nuclear area, plays a role in influencing chromatin state, highlighting the importance of cell-nucleus coordination in gene regulation during cell crowding.

To demonstrate the enhanced predictive power of the nonlinear model, we plotted predicted versus measured H3K27me3 levels, revealing a strong Pearson correlation ($r = 0.720$). To further evaluate prediction robustness, we reconstructed the spatial distribution of H3K27me3 by mapping the cell and nuclear distributions within an MDCK monolayer, with each nucleus color-coded according to its normalized H3K27me3 level (Fig. 5F). The close resemblance between the predicted and measured maps supports the central hypothesis of this study: Despite the inherent heterogeneity of chromatin states, their predictability reveals a fundamental biological principle – that the chromatin landscape is governed by the interplay between cellular and nuclear morphology.

Building on our findings, we investigated how H3K27me3 levels are associated with morphological properties in DN-KASH cells, where nucleus-cytoskeleton linkage is disrupted. Consistent with our earlier observations, LINC disruption significantly reduced the predictive power of nucleus area for H3K27me3 levels (Fig. 5G). However, when all morphological features were incorporated, including NC ratios (Fig. 5H) and nonlinear relationships (Fig. 5I), prediction accuracy was restored to levels comparable to control samples. This restored predictive accuracy suggests that LINC disruption reprograms the relationship between histone mark levels and cellular morphology. For example, in the absence of functional LINC, H3K27me3 levels could still be influenced by cellular features other than nuclear size.

To better understand this reprogramming, we conducted a dropout analysis, systematically removing individual morphological features from the multi-linear model and evaluating their impact on prediction accuracy. Rankings of feature importance revealed distinct trends. In control samples, nucleus area, nucleus shape index, and NC aspect ratio were the strongest predictors of H3K27me3 levels. In contrast, in DN-KASH samples, cell perimeter and circularity became dominant predictors (Fig. 5J). These findings indicate that LINC disruption alters the specific morphological features responsible for the association with H3K27me3 levels rather than abolishing the association altogether.

Corresponding discussion and figures have been added to the manuscript as a subsection in the Results titled “Nuclear area as a primary morphological predictor of H3K27me3 levels”, and as Fig. 5, respectively.

In addition to the three new experiments, we made considerable efforts to explore further investigations suggested by the Reviewer, particularly the proposed time-course experiments. However, these experiments presented significant technical challenges. Our confinement time-course data show that measurable changes in H3K27me3 levels take approximately 8 hours to manifest. During this period, any external physical stimulus could lead to extensive remodeling of cellular organelles, including the nucleus

and cytoskeleton. This dynamic remodeling adds considerable complexity to isolating the specific molecular mechanisms underlying these changes. Additionally, hypotonically shocked cells experience significant stress within 20 minutes, driven by disrupted cell membranes and intracellular osmosis. Overcoming these challenges will require the development of a specialized experimental platform, which lies beyond the scope of this study but is being developed for future research.

Finally, the Reviewer raised the possibility that mechanical stress (pre-stressing) within the epithelial monolayer may influence chromatin state. To address this, we conducted a comprehensive literature review and developed a preliminary experimental plan. Our literature survey revealed that testing this hypothesis would require integrating junctional tension inference with FRET sensor technology into our experimental framework. While this line of investigation holds promise for uncovering novel mechanisms, adapting and validating these advanced tools would entail considerable effort to ensure accurate and reliable results. As our lab currently lacks the necessary expertise and infrastructure for such complex methodologies, we are unable to pursue this direction at present. Nonetheless, we intend to explore this avenue in future studies.